# *TemporalBench*: Evaluating Fine-Grained Temporal Dynamics Understanding for Multimodal Models

## Abstract

Understanding fine-grained temporal dynamics is crucial for multimodal video comprehension and generation. Due to the lack of fine-grained temporal annotations, existing video benchmarks mostly resemble static image benchmarks and are insufficient at evaluating models for temporal understanding. In this paper, we introduce *TemporalBench*, a benchmark dedicated to evaluating **fine-grained temporal understanding** in videos. *TemporalBench* consists of ∼15K video question-answer pairs, derived from ∼2K high-quality human annotations detailing the temporal dynamics. As a result, our benchmark provides a unique testbed for evaluating various temporal understanding and reasoning abilities such as *action frequency, motion magnitude, event order, etc.* Moreover, it enables evaluations on various tasks such as both short and long video understanding, as well as different models including multimodal embedding models and text generation models. Furthermore, we notice a critical pitfall for multi-choice QA where LLMs can detect the subtle changes in negative captions and find a "centralized" description as a cue for its prediction, and we propose Multiple Binary Accuracy (MBA), a new metric for dense temporal understanding, to correct such bias. Results show that state-of-the-art models like Gemini-2.5-Pro achieve only 43.6% question answering accuracy on *TemporalBench* short video QA, demonstrating a significant gap (∼ 24%) between humans and AI in temporal understanding. We hope that *TemporalBench* can foster research on improving models' temporal reasoning capabilities. Both dataset and code will be available.

## 1 Introduction

Videos capture the temporal evolution of visual scenes through sequences of frames. This temporal dimension is fundamental to video understanding—distinguishing it from static image comprehension. Yet recent evaluations reveal a surprising trend: vision-language models trained solely on images often match or exceed the performance of video models on popular benchmarks (Xu et al., 2016; Jang et al., 2017). Even more striking, models processing just a single frame can achieve competitive accuracy on tasks ostensibly requiring temporal reasoning (Wu, 2024; Kim et al., 2024). This single-frame bias suggests that existing benchmarks fail to genuinely evaluate temporal understanding.

The root cause lies in the coarse-grained nature of current video annotations. Consider a typical question-answer pair: "What is the person doing?" with answer "cooking." Such descriptions capture only high-level semantics that are often visible in any individual frame. The temporal dynamics—how the cooking unfolds, the sequence of actions, the precise manipulations—remain unannotated. Without these fine-grained temporal details, benchmarks reduce to static scene recognition tasks where temporal reasoning becomes optional rather than essential.

This limitation has profound implications. Models achieve high scores without learning to process temporal information, creating an illusion of video understanding. The benchmarks meant to drive progress in temporal reasoning instead reward static visual recognition. The gap between benchmark performance and genuine temporal comprehension remains unmeasured and unaddressed.

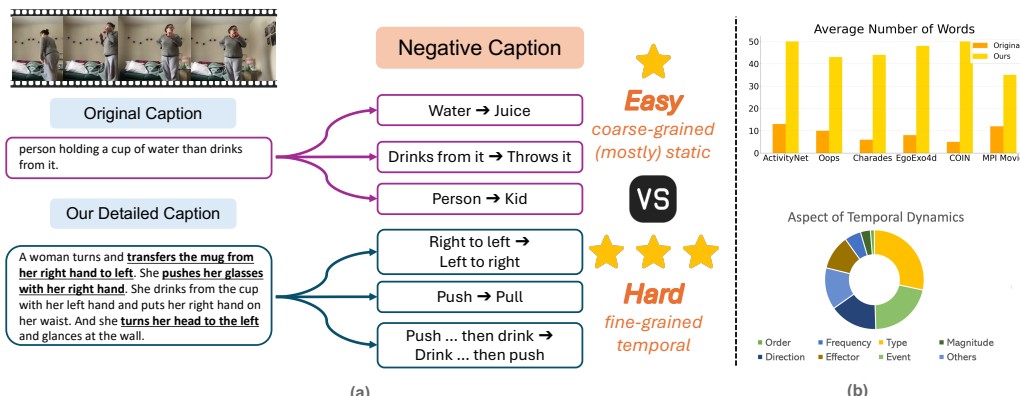

Figure 1: (a) **Comparison between negative captions generated from the original caption and our detailed caption in *TemporalBench*.** With fine-grained details, the negatives are more temporal centric and difficult. (b) *TemporalBench* differs from existing benchmarks by the average number of words per video (left), and the coverage of various temporal aspects (right).

To address this fundamental issue, we introduce *TemporalBench* (Figure 1), a benchmark specifically designed to evaluate fine-grained temporal understanding in videos. Our benchmark consists of ∼**15K** question-answer pairs (10K short, 5K long) derived from ∼**2K** meticulously annotated video captions that capture rich temporal dynamics. Unlike existing benchmarks, our annotations detail the precise sequence of actions, their duration, frequency, and temporal relationships—information that cannot be inferred from a single frame.

The key insight behind *TemporalBench* is simple: by annotating temporal dynamics with fine-grained detail, we create evaluation tasks that inherently require temporal reasoning. When a caption specifies "slices the ginger three times," distinguishing it from "slices the ginger twice" demands processing multiple frames. When describing "transfers the mug from right hand to left, then pushes glasses up," understanding the action sequence requires temporal comprehension. These detailed annotations transform video understanding from optional to essential.

As shown in Figure 2, we collect videos spanning diverse domains: procedural videos (Tang et al., 2019), human activities (Krishna et al., 2017; Gao et al., 2017), ego-centric videos (Grauman et al., 2024), movie clips (Rohrbach et al., 2015), professional gymnastics (Shao et al., 2020), and unexpected events (Epstein et al., 2020). Each video receives detailed temporal annotation through a rigorous two-stage process involving qualified annotators and expert refinement. We then generate challenging negative captions using large language models, creating minimal but semantically significant variations that test precise temporal understanding.

*TemporalBench* offers three defining characteristics:

- **Fine-grained temporal distinctions.** Our negative captions probe specific temporal aspects: action frequency ("twice" vs "three times"), ordering ("then" vs "before"), and dynamics ("pushes" vs "pulls")—details invisible in single frames.

- **Short and long video evaluation.** We test both immediate temporal understanding (<20 seconds) and extended temporal reasoning (<20 minutes) by composing multiple clip descriptions.

- **Unified evaluation framework.** *TemporalBench* supports both discriminative models (XCLIP (Ni et al., 2022), ImageBind (Girdhar et al., 2023)) and generative models (GPT-4o, Gemini), enabling comprehensive assessment across architectures.

During evaluation, we identified a critical flaw in standard multi-choice question answering: when all incorrect options derive from small modifications to the correct answer, models can exploit the "centralized" pattern without understanding content. We address this with Multiple Binary Accuracy (MBA), decomposing each multi-choice question into independent binary decisions that eliminate such shortcuts.

Our results reveal the severity of the temporal understanding gap. State-of-the-art GPT-4o achieves only 38.5% MBA on short videos, compared to 67.9% human performance—a ∼30% deficit. Performance degrades further on long videos, confirming that current models, despite high scores on

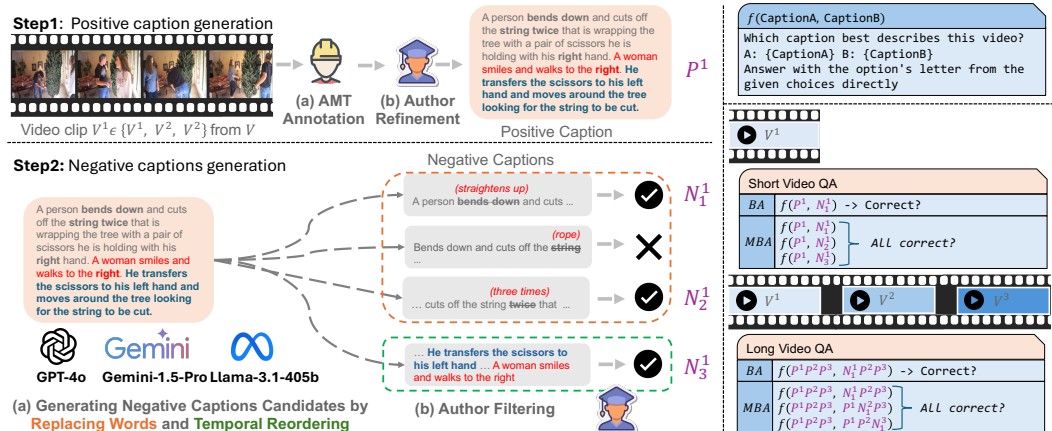

Figure 2: **Overview of the annotation and evaluation pipeline of** *TemporalBench*. **Left**: In step 1, we fist collect high-quality captions for the videos using qualified AMT annotators followed by refining them. In step 2, we leverage existing LLMs to generate negative captions by replacing select words and reordering the sequence of actions before filtering them ourselves. **Right**: *TemporalBench* supports both short and long video QA, where the descriptions of the long video are concatenated from its video clips.

existing benchmarks, lack genuine temporal reasoning capabilities. *TemporalBench* thus provides both a diagnosis of current limitations and a path toward developing models with true temporal understanding.

## 2 RELATED WORK

**Large Multimodal Models.** The proliferation of Large Multimodal Models (LMMs), from proprietary systems like GPT-4V (OpenAI, 2023) and Gemini Gemini Team (2024), to open-source models like LLaVA Liu et al. (2023a) and Qwen-VL series (Bai et al., 2023), has created a need for more challenging evaluation benchmarks. While these models have demonstrated impressive capabilities in image understanding, their ability to reason about fine-grained temporal dynamics remains under-explored.

**Multimodal Understanding Benchmarks.** The recent significant advancements have resulted in more versatile multimodal models, making it imperative to thoroughly and extensively evaluate their visual understanding and reasoning abilities. Conventional multimodal benchmarks like VQA (Antol et al., 2015), GQA (Hudson & Manning, 2019) and VizWiz (Gurari et al., 2018) have been revitalized and used for evaluating the general visual question answering performance for LMMs. Some other question answering benchmarks like TextVQA (Singh et al., 2019), DocVQA (Mathew et al., 2021) and InfoVQA (Mathew et al., 2022) have also been employed to validate the text-oriented understanding. Recent studies have introduced a variety of new benchmarks, such as SEED-Bench (Li et al., 2023a), MMBench (Liu et al., 2023b) and MM-Vet (Yu et al., 2024b) for evaluating the models' integrated problem-solving capabilities, and MMMU (Yue et al., 2024a) and MathVista (Lu et al., 2024) for scientific and mathematical reasoning. In addition, the commonly known hallucination problem also appears in LMMs, and is also investigated in POPE (Li et al., 2023b), MMHal-Bench (Sun et al., 2023) and Object HalBench (Yu et al., 2024a), *etc*.

**Video Understanding Benchmarks.** Recently, an increasing amount of research is transitioning its focus from the image to the video domain Shangguan et al. (2024); Zhou et al. (2018); Kesen et al. (2024); Li et al. (2024d); Chen et al. (2024); Cores et al. (2024). Videos differ from images in that they possess more complex content with temporal dynamics. This unique aspect calls for a different set of metrics and benchmarks. Many efforts have leveraged existing video question answering benchmarks (Xu et al., 2017; Yu et al., 2019b; Xiao et al., 2021; Li et al., 2024d; Shangguan et al., 2024) built on top of video-text datasets (Chen & Dolan, 2011; Xu et al., 2016; Zhang et al., 2019). More recently, several LMM-oriented benchmarks have been proposed for different aspects such as long-form egocentric understanding with EgoSchema (Mangalam et al., 2024), and temporal understanding and ordering like Tempcompass (Liu et al., 2024c). VideoCon Bansal et al. (2024) curated

negative captions using LLM from the original caption, yet lacks fine-grained details shown in Figure 1. MV-Bench (Li et al., 2024c) compiles existing video annotations from different disciplines into a new benchmark, while Video-MME (Fu et al., 2024) and MMWorld (He et al., 2024b) claim to support a comprehensive evaluation of video understanding and world modeling, respectively. YouCook2 Zhou et al. (2018) emphasizes procedure videos but lacks fine-grained temporal understanding. VITATECS Li et al. (2024d), TempCompass Liu et al. (2024c), TOMATO Shangguan et al. (2024) and TVBench Cores et al. (2024) work towards better model temporal dynamics via the counterfactual manner, but still lack the dense captions for fine-grained details.Our *TemporalBench* serves the common goal of evaluating models for video understanding but differs in several aspects. On the one hand, we exhaustively curate videos from different domains and ask human annotators to annotate the visual contents with as much detail as possible. On the other hand, we particularly focus on temporal dynamics such as human actions and human-object interactions that exist exclusively in videos and which are crucial for video understanding, reasoning and forecasting.

## 3 *TemporalBench*

Compared to static images, videos inherently contain significantly more fine-grained temporal information, as they capture the unfolding of actions and events over time. Existing multimodal video understanding benchmarks (Xu et al., 2016) mostly evaluate models' coarse-level understanding of videos. An example from the existing video understanding benchmark is the question, *"What action is happening in the video?"* with the answer, *"Cooking."* However, such

Table 1: Dataset characteristics including number of samples ( # Samples), average number of words in original captions and our fine-grained captions.

| Dataset | # Samples | Org # words | Ours # words |
|---|---|---|---|
| ActivityNet | 281 | 13.03 | 49.55 |
| EgoExo4D | 307 | 7.73 | 47.79 |
| Charades | 298 | 6.21 | 44.16 |
| Movie Description | 326 | 12.39 | 35.33 |
| Oops | 294 | 10.06 | 43.27 |
| COIN | 385 | 5.01 | 50.06 |
| FineGym | 288 | 21.92 | 21.92 |
| *TemporalBench* (ours) | 2179 | 10.91 | 42.02 |

types of coarse-level video questions have been demonstrated to be easily solved with just a single frame (Wu, 2024) or even by a text-only LLM (Tan et al., 2024; Mangalam et al., 2024).

Such phenomena arises due to a fundamental limitation in text descriptions in those benchmarks. As a result of their coarseness, the positive and negative options for video question-answering can usually be distinguished without understanding the temporal dynamics. For example, in Figure 1, models only need to choose between *"The man is cooking"* and *"The man is exercising"*, which can be answered without understanding the temporal aspects.

To address this limitation, we carefully design a human annotation pipeline to curate highly detailed descriptions about the activities in the videos. Given the detailed video clip descriptions, such as *A right hand holds a piece of peeled ginger while a knife is held in the left and makes 3 slices off the ginger.*, the negative captions can be curated to truly reflect whether a model understands the temporal dynamics, such as changing *"three slices"* into *"two slices"*. In a nutshell, such highly detailed temporal annotations can be used to carefully examine whether a multimodel video model truly understands the temporal state transition in videos.

Our benchmark enriches several fundamental video understanding tasks due to its detailed captions:

- **Fine-grained video question answering.** Given a detailed positive caption, multimodal video models need to distinguish it from the associated negative where a slight modification is made to temporal descriptions, *e.g., "push the eyeglasses up"* versus *"pull the eyeglasses down"*, or *"cut 3 slices off"* versus *"cut 2 slices off"*.

- **Long video understanding with fine-grained activity inspection.** Since the video clips are extracted from a long source video, the respective video clip descriptions can be concatenated to form a longer video description which can be pivoted to the long video understanding task, where we find that all current multimodal video models suffer.

- **Dense video-text matching and retrieval.** Our detailed video captions can be naturally employed to evaluate video-language embedding models such as XCLIP (Ni et al., 2022). Given a positive caption and several negative captions, we can evaluate whether CLIP (Radford et al., 2021) based

Table 2: Comparsion between *TemporalBench* and other video understanding benchmarks.# Words/s denote the average number of words per second, and A & M denotes Auto & Manual during the annotation process.

| Dataset | Num QA | # Video | Open-domain | Annotation | Avg Sec | # Words/s |
|---|---|---|---|---|---|---|
| MVBench | 4,000 | 3,641 | ✓ | Auto | 16 | 0.22 |
| EgoSchema | 5,000 | 5,063 | ✗ | A & M | 180 | 0.12 |
| LongVideoBench | 6,678 | 3,763 | ✓ | Manual | 473 | 0.16 |
| Video-MME | 2,700 | 900 | ✓ | Manual | 1018 | 0.006 |
| SEED-Bench | 3,757 | 2,320 | ✗ | A & M | 110 | 0.97 |
| MMWord | 6,627 | 1,910 | ✓ | A & M | 102 | 0.56 |
| *TemporalBench* (short) | 9,867 | 2,179 | ✓ | A & M | 8 | 6.27 |
| *TemporalBench* (long) | 5,485 | 1,574 | ✓ | A & M | 123 | 2.62 |

video embedding models can distinguish the subtle differences in captions. In addition, given a set of positive video-text pairs, video retrieval performance can be evaluated, similar to image retrieval on COCO (Lin et al., 2014) and Flickr30K (Young et al., 2014).

Our details video captions support tasks like fine-grained video captioning shown in the Supp., and enable future tasks such as video grounding from detailed text descriptions and text-to-video generation with detailed prompts.

### 3.1 VIDEO COLLECTION

We collect video clips from a wide range of sources across *diverse* domains, where the majority comes from existing video grounding dataset test/val split. Our dataset includes a wide spectrum of video types from seven sources, including (1) procedure videos *e.g.,* COIN (Tang et al., 2019), (2) human activities *e.g.,* ActivityNet-Captions (Yu et al., 2019a) and Charades (Krishna et al., 2017), (3) ego-centric videos *e.g.,* EgoExo4D (Grauman et al., 2024), (4) movie descriptions (Rohrbach et al., 2015), (5) professional gymnasium videos *e.g.,* FineGym (Shao et al., 2020), and (6) unexpected humor videos Oops (Epstein et al., 2020). We sample around 300 video clips from the *validation and test sets* of each video dataset, which results in 2K videos. The statistics of *TemporalBench* is shown in Table 1, comparsion with existing benchmarks are shown in Table 2 and also Table 6 in the appendix.

We intentionally filter out video clips that (1) are mostly static by leveraging optical flow (Farnebäck, 2003), (2) contain multiple scene transitions by detecting shot changes in videos and (3) last longer than 20 seconds. We observe that the large amount of information in long videos make it difficult for annotators to provide detailed action descriptions. The distribution of video lengths is shown in the Supp. Additionally, we remove the audio from the videos during annotation to ensure that all informative signals come solely from the visual frames, preventing the answers from being influenced by the audio.

### 3.2 VIDEO CAPTION ANNOTATION PROCESS

**Positive Captions Annotation.** We employ a two-stage human labeling process for curating video captions with fine-grained activity descriptions, where the qualified Amazon Mechanical Turk (AMT) workers are first instructed to give a detailed video caption. Then, authors of this work refine the caption by correcting the mistakes and adding missing details *w.r.t.* the actions. The overall pipeline is shown in Figure 2. All video clips are annotated following the same pipeline except for Finegym (Shao et al., 2020) as it has already provided accurate and detailed action descriptions for professional gymnasium videos. Consequently, we reuse its annotations.

We first use 3 probing video captioning questions with 2 in-context examples as the onboarding task for AMT master workers, where prompts are shown in the Supp. We manually inspect the soundness and amount of temporal details of the AMT worker captions to select high quality AMT video captioning workers. During the annotation process by AMT workers, we also continue to remove the unqualified workers based on the ratio of the captions that authors in this paper refined. In this way, we ensure that the AMT provides a high quality initial point for positive captions.

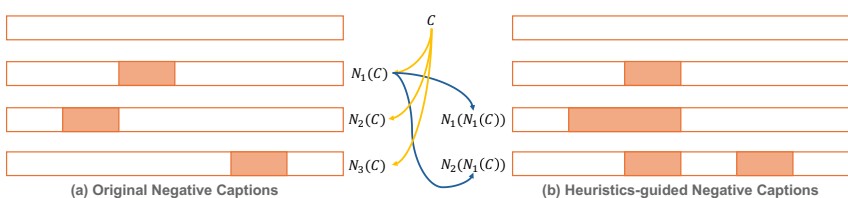

Figure 3: An illustration of multi-choice QA with (a) original and (b) heuristics-guided negative captions. Orange blocks indicate the altered contents from the positive option (green box).

**Negative Caption Annotation.** Our negative captions are aimed at confusing multimodal video models with respect to fine-grained activity details, such as changing *"cut a ginger twice using a knife"* to *"cut a ginger three times using a knife"*. We construct negatives upon two granularities: word level and event level. Specifically, word level negatives denote the case where a certain word or phrase is replaced while event level negatives denote the case where the order of two events are reversed. Empirically, we find that LLMs can produce more creative and diverse negatives compared to AMT workers and authors. Therefore, we leverage three leading LLMs, GPT-4o (OpenAI, 2024), Gemini-1.5-Pro (Gemini Team, 2024) and Llama-3.1-405b (Dubey et al., 2024) to curate a diverse set of negative caption candidates instructed by 3 in-context examples, with up to 9 negatives at word level and 6 negatives at event level. Prompts are shown in the Supp.

Afterwards, the authors of this work review those negative caption candidates in the format of multi-choice QA. Essentially, we build a reliable evaluation pipeline w.r.t. the detailed captioning, alleviating the limitation of previous This process results in our complete *TemporalBench* dataset with ~2K high-quality human-annotated video captions and ~10K (short) video question-answer pairs.

### 3.3 A Pitfall in Multi-choice Question Answering

Multi-choice QA is standard for evaluating LMMs (MMMU (Yue et al., 2024a), MathVista (Lu et al., 2024), EgoSchema (Mangalam et al., 2024)). However, recent studies (Cai et al., 2024; Yue et al., 2024b) show that pure LLMs often match or exceed LMM performance without visual input. This is attributed to (1) poorly designed questions answerable without the video, or (2) data contamination.

We identify another critical pitfall. If all negatives are slight modifications of the correct answer, LLMs can detect a "centralized" description as a cue (Figure 3). To test this, given positive caption $C$ and negative $N(C)$, we derive further negatives from $N(C)$ (e.g., $N_1(N(C))$, $N_2(N(C))$). With options $[C, N(C), N_1(N(C)), N_2(N(C))]$, $N(C)$ becomes "centralized". Surprisingly, text-only GPT-4o selects $N(C)$ 66.4% of the time, and $C$ only 6.4%. This aligns with psychology findings (Furman & Wang, 2008) where humans exploit similarity cues in multiple choice.

Motivated by this, we propose decomposing multi-choice QA into multiple binary QAs, eliminating the "centralized option." We introduce Multiple Binary Accuracy (MBA), which reflects the compounded difficulty of answering *all* sub-questions: $\text{MBA} = \prod_{j=1}^{M} \text{acc}_j$, where $\text{acc}_j$ is the correctness of sub-question $j$ (e.g., $92\%^5 = 65.9\%$). Given $M$ negatives, random chance drops from $\frac{1}{M+1}$ (multi-choice) to $(\frac{1}{2})^M$ (MBA). Since $(\frac{1}{2})^M < \frac{1}{M+1}$ for $M > 1$, MBA is a significantly more challenging metric.

## 4 Experiments

### 4.1 Experiment Setup

We evaluate both (1) multimodal video text generation models, including GPT-4o, o1, o4-mini (OpenAI, 2024), Gemini-2.5-Pro (Gemini Team, 2024), Claude-3.5-Sonnet (Anthropic, 2024), Qwen2.5VL, Qwen2VL (Wang et al., 2024), LLaVA-OneVision (Li et al., 2024a), LLaVA-Next-Video (Zhang et al., 2024b), MA-LMM (He et al., 2024a), VideoLLaVA (Lin et al., 2023), InternLM-Xcomposer-2.5 (Zhang et al., 2024a), Matryoshka Multimodal Models (Cai et al., 2024), and (2) multimodal video embedding models, including XCLIP (Ni et al., 2022), ImageBind (Girdhar et al., 2023), and LanguageBind (Zhu et al., 2024). We exponentially increase the number of frames to study its effect on video understanding. *TemporalBench* takes comparable inference com-

Table 3: *TemporalBench* performance of various mmodels under the binary QA accuracy (BA) and multiple binary QA settings (MBA) for short videos. The prefix "T-" indicates MBA performance for the annotated subset in our *TemporalBench*. We show the result with the best average MBA performance for each model with respect to the number of frames, denoted as # Frames.

| Model | # Frames | T-ActivityNet | T-Charades | T-FineGym | T-Movie | T-Oops | T-COIN | T-EgoExo4D | BA | MBA |
|---|---|---|---|---|---|---|---|---|---|---|
| Human Performance | - | **68.7** | **82.2** | **36.1** | **74.2** | **69.7** | **70.6** | **71.0** | **89.7** | **67.9** |
| Random Chance | - | 11.1 | 13.8 | 6.2 | 12.1 | 5.6 | 11.2 | 5.6 | 50.0 | 9.5 |
| *Video Embedding Models: Text + Multiple Frames as Input* | | | | | | | | | | |
| XCLIP | 8 | 14.2 | 16.1 | 7.3 | 19.9 | 8.8 | 15.6 | 6.8 | 51.6 | 12.9 |
| ImageBind | 2 | 17.4 | 16.8 | 7.3 | 19.0 | 11.2 | 16.1 | 9.1 | 53.0 | 14.0 |
| LanguageBind | 8 | 22.4 | 15.1 | 6.6 | 19.3 | 10.9 | 15.6 | 11.1 | 52.8 | 14.5 |
| *Video Multimodal Generative Models : Text + Multiple Frames as Input* | | | | | | | | | | |
| Gemini-2.5-Pro | 1FPS | 81.1 | 78.0 | 68.7 | 81.9 | 81.5 | 78.8 | 81.5 | 78.7 | 43.6 |
| OpenAI o1 | 32 | 82.0 | 75.9 | 70.2 | 80.6 | 79.7 | 80.0 | 82.1 | 78.6 | 43.0 |
| OpenAI o4-mini | 32 | 78.7 | 74.5 | 71.0 | 78.7 | 79.6 | 78.4 | 81.7 | 77.5 | 40.7 |
| Gemini-2.5-Flash | 1FPS | 78.4 | 71.8 | 64.0 | 78.4 | 79.2 | 78.0 | 78.6 | 75.5 | 36.7 |
| GPT-4o | 16 | 48.8 | 42.6 | 18.8 | 41.7 | 31.6 | 46.5 | 36.5 | 75.7 | 38.5 |
| Gemini-1.5-Pro | 1FPS | 34.9 | 24.5 | 8.3 | 35.6 | 22.8 | 34.3 | 21.8 | 67.5 | 26.6 |
| Claude-3.5-Sonnet | 8 | 29.9 | 27.5 | 11.1 | 28.2 | 16.3 | 29.6 | 20.5 | 65.5 | 23.6 |
| Qwen2-VL-72B | 32 | 43.8 | 42.6 | 16.7 | 45.1 | 36.7 | 43.6 | 37.1 | 75.8 | 38.3 |
| Qwen2.5-VL-7B | 1 FPS | 71.3 | 65.7 | 56.0 | 71.0 | 69.1 | 68.0 | 70.0 | 67.2 | 26.1 |
| Qwen2-VL-7B | 32 | 32.4 | 32.2 | 4.9 | 35.9 | 18.4 | 25.5 | 21.8 | 64.4 | 24.7 |
| LLaVA-OneVision-72B | 8 | 45.2 | 36.2 | 11.8 | 41.1 | 31.0 | 34.5 | 30.3 | 72.1 | 33.0 |
| LLaVA-NeXT-Video-34B | 32 | 30.6 | 26.8 | 10.4 | 24.8 | 18.0 | 25.2 | 17.3 | 64.0 | 22.0 |
| MA-LMM | 4 | 12.5 | 16.4 | 3.5 | 11.0 | 5.1 | 11.4 | 4.9 | 48.0 | 9.4 |
| $M^3$ | 6 | 21.0 | 20.1 | 6.6 | 19.6 | 10.2 | 15.1 | 10.4 | 56.4 | 14.8 |

pute as existing video understanding benchmarks such as NextQA Xiao et al. (2021). Details are shown in the Supp.

To study the effect of single frame bias and text bias, we also evaluate models trained on single images, including LLaVA-1.5 (Liu et al., 2024a), LLaVA-NeXT (Liu et al., 2024b), and Phi-3V (Abdin et al., 2024). In the latter case, we evaluate the LLMs including GPT-4o (OpenAI, 2024), Gemini-1.5-Pro (Gemini Team, 2024), Yi-34B (Young et al., 2024), Vicuna (Chiang et al., 2023) and Flan-T5 (Wei et al., 2021) without using videos at all.

## 4.2 HUMAN PERFORMANCE

We use Amazon Mechanical Turk to evaluate human performance. Note that we exclude the positive caption annotators to ensure that there is no data contamination. Again, we use an onboarding test using a held out binary video QA evaluation set which has clear answers.

## 4.3 FINE-GRAINED VIDEO QUESTION ANSWERING ON SHORT VIDEOS

The results for multimodal generative models and embedding models are shown in Table 3. Note that we show the result with the best average multiple binary QA (MBA) performance for each model with respect to the number of frames. Results under different frames can be found in the Supp. Several interesting findings arise:

**The performance of any video model is far from human performance.** As shown in Table 3, humans show an average performance of 67.9% on MBA, which is significantly higher than the best models, Gemini 2.5 Pro, by ∼24%. Therefore, there is a large gap between models' performance and human performance. Note that we are employing standard AMT workers instead of domain experts, meaning that the expert-level accuracy can be even higher, especially for professional videos like FineGym.

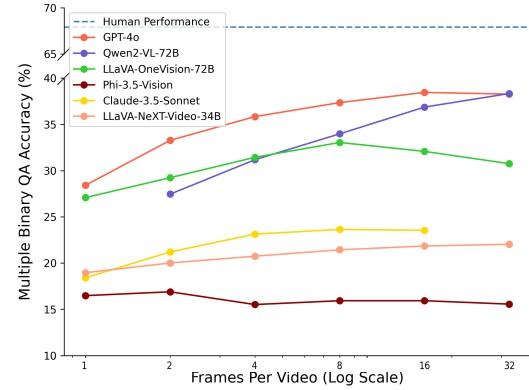

Figure 4: **Model performance on *Temporal-Bench*** with varying numbers of frames for short video understanding.

**Models show limited performance gains with more frames**. Shown in Figure 4, with more frames, multimodal video models usually show better performance. However, performance generally saturates around 8-16 frames, meaning that models struggle to improve fine-grained activity understanding even with more frames. This is a clear contrast with human performance, showing that there is a large space for multimodal video models to improve.

**Multiple Binary QA is a more challenging metric.** Multiple Binary QA, as proposed in Section 3.3, prevents a model from exploiting cues in the answer choices, and evaluates whether a model truly understands the temporal dynamics in the video by splitting a single $M + 1$-way multiple choice question into $M$ binary choice questions. For example, Gemini-2.5-Pro receives 78.7% accuracy but only 43.6% on multiple binary accuracy, showing a huge gap. These results indicate that understanding the fine-grained temporal dynamics is still a challenging task for current proprietary models and open-sourced models.

**Video embedding models show near chance performance.** All multimodal video embedding models, including XCLIP, LanguageBind, and ImageBind show near random chance performance. One reason could be that their small embedding size (typically a vector of size 768-2048) is insufficient to capture fine-grained temporal details.

## 4.4 LONG VIDEO UNDERSTANDING

Since our benchmark is annotated at the video clip level, we can easily extend it to long video understanding by concatenating the captions of different video clips within the same original video. In our study, we choose video datasets from AcitivityNet, Charades, EgoExo4D, COIN and FineGym. We randomly sample video clips within the same original video, and then crop a new video segment whose starting time corresponds to that of the earliest sampled video clip and whose ending time corresponds to that of the latest sampled video clip. We then concatenate all the sampled video captions together to form a single long detailed description corresponding to the new video segment. Given this positive caption, we generate negative captions for it by replacing the positive caption of one of the sampled video clips with its negatives. The model is then tasked to choose the correct long caption out of multiple choices. We control the random chance multiple binary QA performance to be ∼9.5%, resulting in an apple-to-apple comparsion with in Sec 4.3. In this way, we investigate whether multimodal video models can understand and distinguish fine details in a long video. Note that each caption for the short video is highly dense and informative, thus can be uniquely matched to its corresponding clip in the long video. Finally, we sampled 1,574 videos with durations ranging between $(0, 20]$ minutes, shown in the Supp.

We show in Figure 6 and table in the Supp., that all multimodal video models show a significant performance drop for this task compared to short video understanding. This is also reflected in all models performing better on relatively shorter videos (*e.g.,* Charades) compared to longer videos (*e.g.,* FineGym). These results indicate that finding the subtle temporal dynamic differences in a long video is indeed an extremely difficult task. It is similar in nature to the needle-in-the-sea task (Kamradt, 2023) in NLP except in the temporal domain. We hope that *TemporalBench* for long video understanding can serve as a very challenging task for future video understanding model development.

## 5 IN-DEPTH ANALYSIS

**Why multiple binary QA instead of multi-choice QA?** As discussed in Section 3.3, in the standard multi-choice QA setting, if negatives are all slightly variations of the positive caption, we find that LLMs can determine the "centralized" caption, and take a shortcut to achieve better performance. To demonstrate

Table 4: Effect of the "Centralized" Caption on text-only GPT-4o.

| Percentage of Predictions Aligned with ⟶ | $C$ | $N_1(C)$ |
|---|---|---|
| $[C, N_1(C), N_2(C)), N_3(C)]$ | **83.3** | 6.4 |
| $[C, N_1(C), N_1(N_1(C)), N_2(N_1(C))]$ | 17.7 | **66.4** |

this, based on one negative caption $N(C)$ in *TemporalBench*, we intentionally generate two negative captions derived from $N(C)$ (instead of $C$), resulting in $N_1(N(C))$ and $N_2(N(C))$. Given two set of options $[C, N_1(C), N_2(C)), N_3(C)]$ and $[C, N_1(C), N_1(N_1(C)), N_2(N_1(C))]$ shown in Figure 3, text-only GPT-4o displays different behaviors. As shown in Table 4, under the intentionally

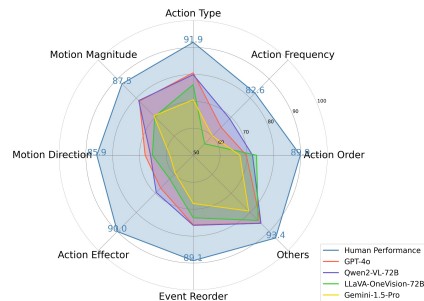

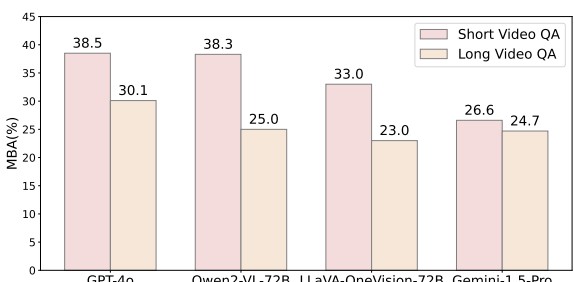

Figure 5: Per-category binary QA accuracy (BA) of various video models on short video QA.

Figure 6: Models' MBA performance on short and long video QA.

designed negative options, GPT-4o will choose $N_1(C)$ under 66.4% cases. This again demonstrates the necessity and advantage of our multiple binary QA accuracy (MBA) metric design over the standard multi-choice QA setting.

**Performance on categories.** Broadly, *TemporalBench* evaluates word level replacement and event level re-ordering. Here we further breakdown the word level replacement into following categories: 1) Action Order (change the order); 2) Action Frequency (1 times *v.s.* two times); 3) Action type (put *v.s.* pull); 4) Motion Magnitude (slightly *v.s.* intensively); 5) Motion Direction/Orientation (forward *v.s.* backward, circular *v.s.* back-and-forth). 6) Action Effector (cutting with left hand *v.s.* cutting with right hand) 7) Others. We prompt GPT-4o to perform 7-way classification, where the statistics is shown in Table 5 in the appendix. Results in

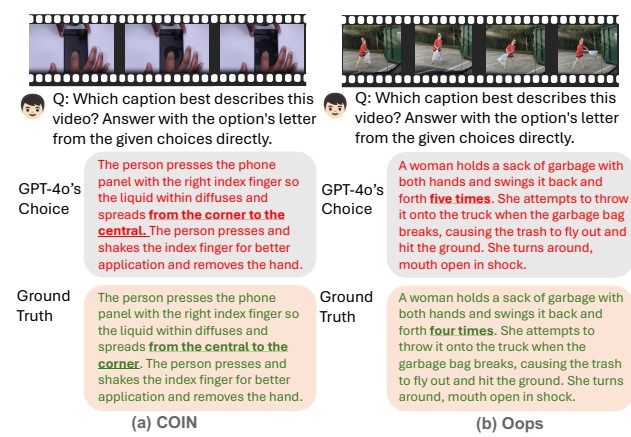

Figure 7: **GPT-4o failure cases in *TemporalBench***: not understanding fine-grained details like motion direction, action frequency, *etc.*.

Fig. 5 indicate that multimodal video models show better performance on "others" category rather than the other categories related to actions. Among the seven categories, models struggle most on action frequency (counting), which shows that they do not memorize repeated occurrences well. Results indicate that emphasizing on fine-grained temporal aspects is critical for the development of multimodal video models, especially from the training data perspective. The visualizations of failure cases in GPT-4o are shown in Fig. 7.

## 6 CONCLUSION AND FUTURE WORK

We propose *TemporalBench*, a novel video understanding benchmark, to evaluate fine-grained temporal understanding abilities of multimodal video models. The video captions in our benchmark are significantly denser than those in existing datasets such as MSRVTT and TGIF, offering detailed temporal annotations. *TemporalBench* also provides a more challenging set of tasks that push current multimodal models beyond coarse-level understanding. The empirical results reveal a substantial gap between human performance and current state-of-the-art models. We also found a critical pitfall for multi- choice QA, where we devise multiple binary accuracy (MBA) to address thi issue. We hope that this benchmark fosters further research in developing models with enhanced temporal reasoning capabilities. Our benchmark could also be easily utilized for other fundamental video tasks such as spatio-temporal localization and text-to-video generation with fine-grained prompts.

REPRODUCIBILITY STATEMENT

We visualize part of the dataset in the submission's supplementary materials. We will also publicly release it along with the code used to evaluate the LMMs upon the paper's acceptance.

ETHICS STATEMENT

This research primarily utilizes publicly available video datasets, which have been collected and annotated by qualified annotators and authors, ensuring compliance with ethical standards. We have made every effort to ensure that the data used respects privacy and contains no personally identifiable information. Furthermore, we acknowledge the potential implications of fine-grained video understanding, especially in sensitive applications such as surveillance and autonomous systems. As such, we advocate for responsible and ethical use of this research, urging caution in deploying these models in real-world scenarios to avoid harmful or unintended consequences.

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

# A    MORE VISUALIZATIONS OF OUR BENCHMARK

In this section, we present comprehensive visualizations of our fine-grained annotations with both positive and negative descriptions. For each benchmark mentioned in Table 1 in the main paper, we provide one video example with its positive annotation and one of the corresponding negative descriptions (there are more than one negative for a single video in our dataset) in Figures 8 & 9. The video examples (*a - g*) are displayed in the same order as their sources in Table 1 in the main paper (7 in total).

Table 5: *TemporalBench* statistics on negative caption types.

| Action Order | Action Frequency | Action Type | Motion Magnitude | Motion Direction | Action Effector | Event Reorder | Others | Overall |
|---|---|---|---|---|---|---|---|---|
| 129 | 530 | 2,802 | 320 | 1,536 | 1,109 | 2,099 | 1,342 | 9,867 |

# B    MORE TASKS THAT CAN BE SUPPORTED BY *TemporalBench*

## B.1    VIDEO CAPTIONING

Our detailed video captions also enables analyzing a model's fine-grained video captioning capabilities. For this, we prompt multimodal video models to generate a caption for an input video, with

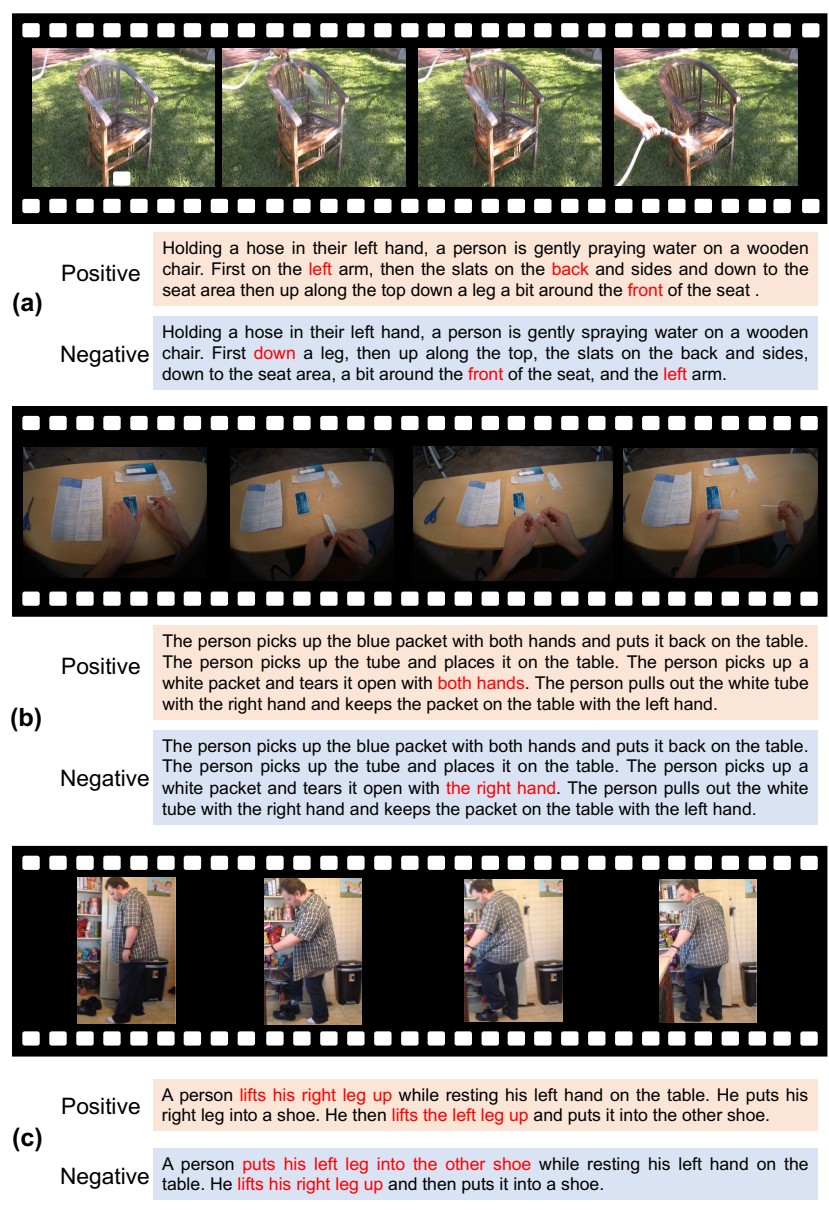

Figure 8: Visualizations (I) of our fine-grained annotations of the videos with both positive and negative descriptions.

Table 6: Comparison of video understanding benchmarks upon annotation type and density.

| Benchmark | Annotation Type | Annotators | Fine-Grained | # Words/Sec* |
|---|---|---|---|---|
| TempCompass | Caption | LLM+Human | Some | 2.12 |
| YouCook2 | Instructions | Human | Procedural | 0.44 |
| VITATECS | Caption | LLM+Human | Yes | 2.48 |
| TOMATO | QA Pairs | Human | Some | 1.67 |
| TVBench | QA Pairs | Parse+Human | Some | 0.07 |
| *TemporalBench* | **Captions** | **Human** | **Yes** | **6.27** |

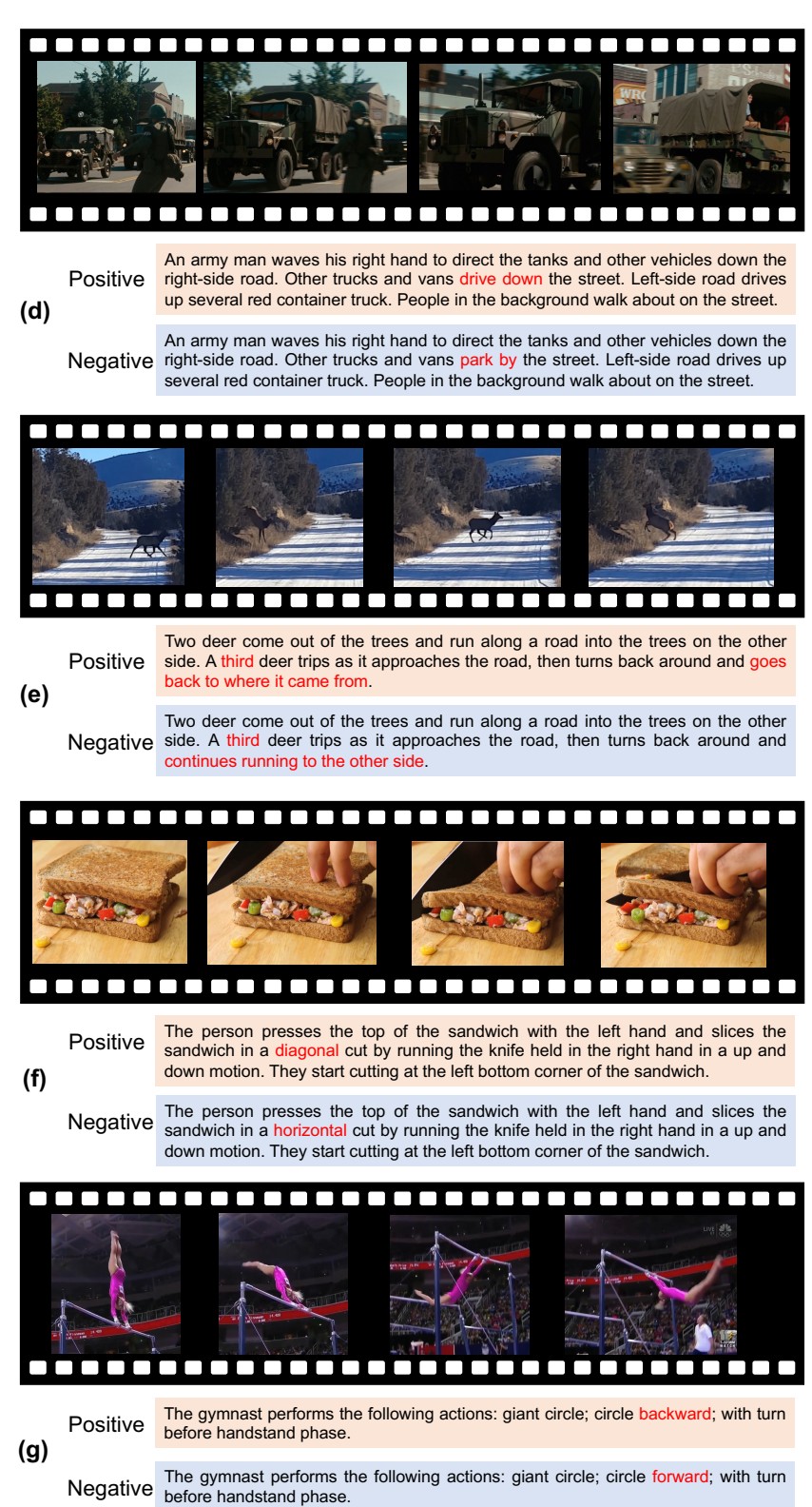

Figure 9: Visualizations (II) of our fine-grained annotations of the videos with both positive and negative descriptions.

Table 7: Comparison of models for video captioning using Caption Similarity, CIDEr, BLEU, and ROUGE metrics. Cosine similarity using sentence transformer reflects the captioning quality the best.

| Model | Similarity | CIDEr | ROUGE | BLEU_1 | BLEU_2 | BLEU_3 | BLEU_4 |
|---|---|---|---|---|---|---|---|
| **Video Multimodal Generative Models : Text + Multiple Frames as Input** | | | | | | | |
| GPT-4o | **61.3** | 7.3 | **19.6** | 24.1 | **11.8** | **5.8** | **3.0** |
| Gemini-1.5-Pro | 56.5 | **10.9** | 19.1 | 19.0 | 9.2 | 4.5 | 2.4 |
| Claude-3.5-Sonnet | 54.1 | 8.6 | 17.1 | 24.4 | 10.3 | 4.4 | 2.1 |
| Qwen2-VL-72B | 56.1 | 9.3 | 19.1 | 15.7 | 8.0 | 4.1 | 2.2 |
| Qwen2-VL-7B | 51.9 | 6.9 | 18.0 | 12.5 | 6.1 | 3.0 | 1.6 |
| LLaVA-OneVision-72B | 55.0 | 9.7 | 18.7 | 23.7 | 11.3 | 5.6 | 2.9 |
| LLaVA-OneVision-7B | 50.1 | 0.3 | 14.5 | 11.1 | 5.1 | 2.2 | 1.1 |
| LLaVA-NeXT-Video-34B | 53.1 | 5.3 | 15.9 | 21.4 | 9.2 | 4.0 | 1.8 |
| LLaVA-NeXT-Video-7B | 50.1 | 2.3 | 15.8 | 18.1 | 7.0 | 2.6 | 1.1 |
| InternLM-XC2.5 | 52.4 | 2.3 | 15.9 | 17.8 | 7.1 | 2.8 | 1.2 |
| VideoLLaVA | 46.0 | 4.5 | 16.9 | 12.6 | 5.4 | 2.3 | 1.0 |
| MiniCPM-V2.6 | 47.2 | 1.5 | 14.2 | 15.5 | 5.4 | 1.9 | 0.8 |
| Phi-3.5-Vision | 42.9 | 3.7 | 16.5 | 20.4 | 8.4 | 3.4 | 1.6 |
| MA-LMM | 38.7 | 3.1 | 15.0 | 10.1 | 4.8 | 2.2 | 1.1 |
| $M^3$ | 47.8 | 3.0 | 16.4 | 16.7 | 6.9 | 2.8 | 1.2 |
| **Large Multimodal Models (LMMs): Text + 1 Frame as Input** | | | | | | | |
| GPT-4o | 52.3 | 7.3 | 17.1 | **25.1** | 11.1 | 5.0 | 2.4 |
| LLaVA-1.5-13B | 47.9 | 4.9 | 18.0 | 22.6 | 9.8 | 4.2 | 2.0 |
| LLaVA-1.5-7B | 45.7 | 6.9 | 17.8 | 22.0 | 9.5 | 4.2 | 2.0 |
| LLaVA-NeXT-34B | 49.1 | 6.2 | 16.7 | 24.2 | 10.4 | 4.6 | 2.2 |
| Phi-3-Vision | 42.0 | 4.0 | 16.1 | 19.9 | 8.3 | 3.4 | 1.6 |

3 captioning examples in the prompt as guidance to mimic the style of our detailed video captions. Note that we remove the FineGym captions due to its different structure compared to other video captions, resulting in 1891 samples. We evaluate the resulting video captioning performance using classical image captioning metrics, CIDEr (Vedantam et al., 2015), BLEU (Papineni et al., 2002) at different n-gram levels, ROUGE (Lin, 2004), as well as the embedding similarity with sentence transformer (Reimers & Gurevych, 2019) between the ground truth caption and the generated caption. Note that we for each model, we use the same number of frames as in Section 4.3 in the main paper.

Results in Table 7 show that GPT-4o achieves the best performance. Interestingly, the results indicate that the embedding similarity aligns most closely with the video QA task results from Sec 4.3 in the main paper. Other classical captioning metrics show inconsistent results. For example, GPT-4o obtains similar performance with one compared to 64 frames on both CIDEr and BLEU scores (e.g., for BLEU_1 24.1 vs. 25.1). On the other hand, all models show similar ROUGE scores. Thus, for the zero-shot captioning task, our findings indicate that text embedding similarity may be the most reliable metric.

## B.2 FUTURE TASKS

Due to the detailed annotation of *TemporalBench*, more tasks can be supported.

- **Video grounding from detailed text descriptions.** Since the video clips are cropped from the source video, with the documented starting and ending time, our benchmark can serve as a fine-grained moment localizing benchmark from text descriptions. This is different from existing video grounding datasets such as Charades-STA (Gao et al., 2017), COIN (Tang et al., 2019), Ego4D (Grauman et al., 2024) where the text descriptions are usually very short, possibly resulting in low temporal localization performance due to the vague and coarse descriptions.

- **Text-to-Video (T2V) generation with detailed prompts.** Given our highly detailed description, a T2V generation model can be evaluated by verifying if the generated videos reflect the fine-grained action details.

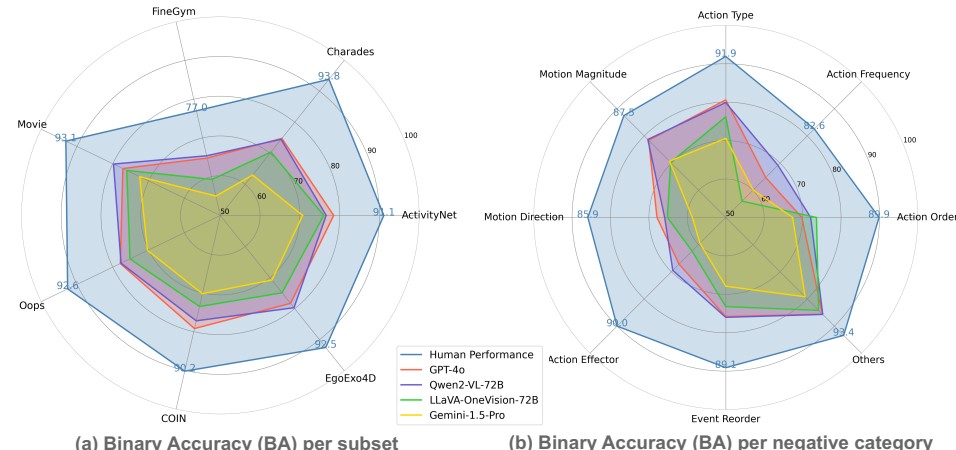

(a) Binary Accuracy (BA) per subset    (b) Binary Accuracy (BA) per negative category

Figure 10: Visualization of binary accuracy for short video QA per (a) subset and (b) negative type. Human performance is much better than GPT-4o, Qwen2-VL-72B, LLaVA-OneVision-72B, and Gemini-1.5-Pro.

## C EXTENDED COMPARISON OF NEGATIVE CAPTIONS

Here we show the extended comparison for the negative captions generated from the original captions and our detailed captions in *TemporalBench* in Figure 11. With fine-grained details, the negatives are more difficult and temporal centric.

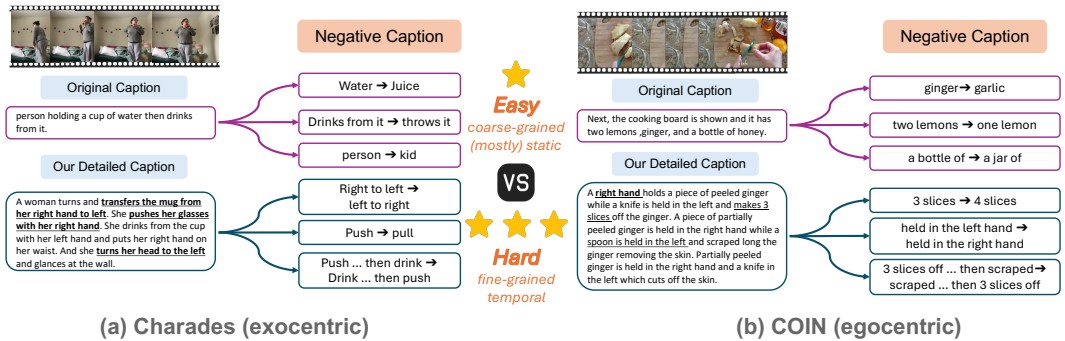

(a) Charades (exocentric)    (b) COIN (egocentric)

Figure 11: **Comparison of negative captions generated from the original captions and our detailed captions in *TemporalBench*.** With fine-grained details, the negatives are more difficult and temporal centric.

## D *TemporalBench* PERFORMANCE UNDER EACH CATEGORY

Multimodal video models' performance under each category is shown in Table 8 and Figure 10 (b). Results indicate that multimodal video models show better performance on "others" category rather than the other categories related to actions. Among the seven categories, models struggle most on action frequency (counting), which shows that they do not memorize repeated occurrences well. Results indicate that emphasizing on fine-grained temporal aspects is critical for the development of multimodal video models, especially from the training data perspective. The visualizations of failure cases in GPT-4o are shown in Figure 12.

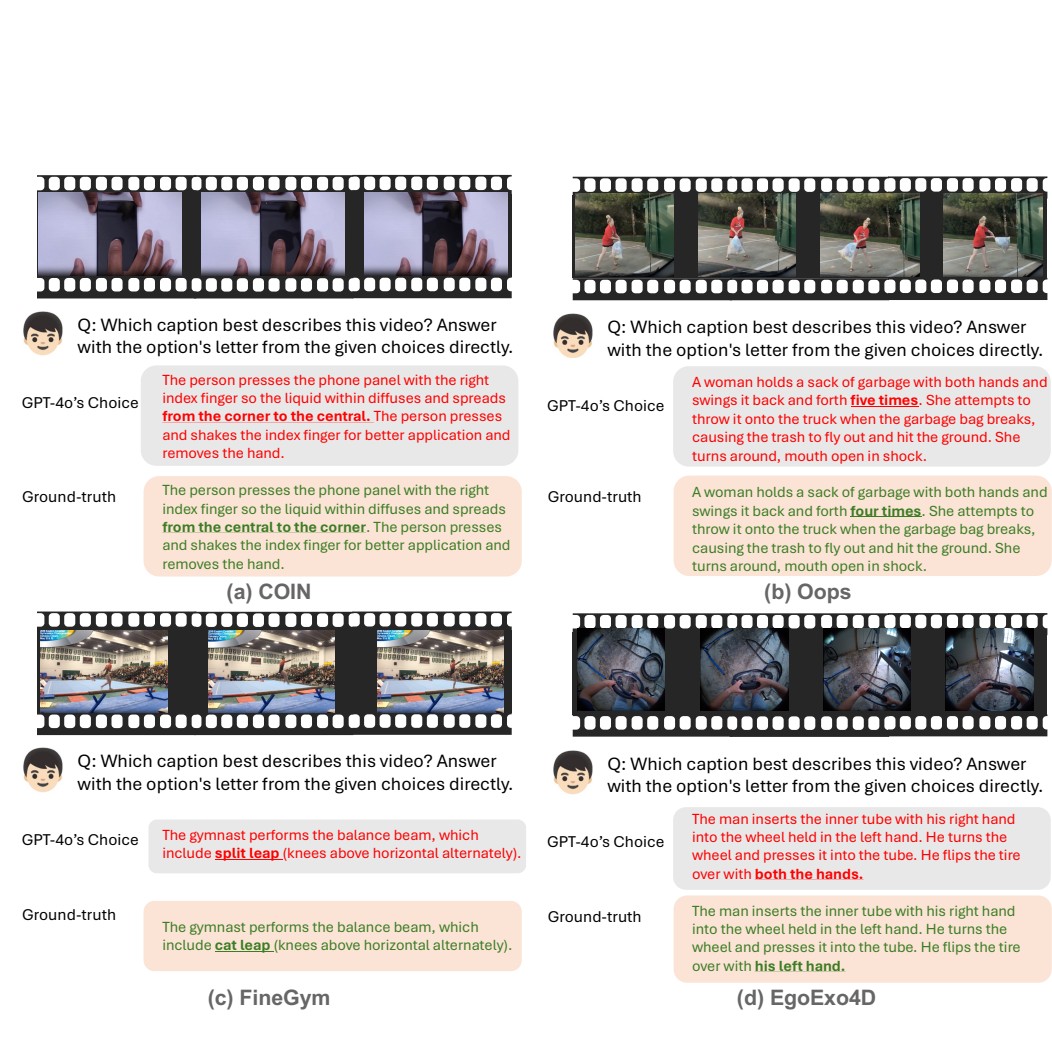

Figure 12: **The failure cases of GPT-4o in** *TemporalBench*. GPT-4o does not understand the fine-grained details well, including motion direction, action frequency, action type, and motion direction.

Table 8: *TemporalBench* short QA performance under each category under BA. Multimodal videos models struggle on certain tasks such as action frequency. We show the result with the best average MBA performance for each model with respect to the number of frames.

| Model | The Number of Frames | Action Order | Action frequency | Action Type | Motion Magnitude | Motion Direction | Action Effector | Event Reorder | Others | Average |
|---|---|---|---|---|---|---|---|---|---|---|
| Human Performance | - | **89.9** | **82.6** | **91.9** | **87.5** | **85.9** | **90.0** | **89.1** | **93.4** | **89.7** |
| Random Chance | - | 50.0 | 50.0 | 50.0 | 50.0 | 50.0 | 50.0 | 50.0 | 50.0 | 50.0 |
| *Video Embedding Models: Text + Multi-Frames as Input* | | | | | | | | | | |
| XCLIP | 8 | 46.5 | 50.8 | 50.9 | 56.9 | 51.2 | 51.7 | 50.1 | 55.6 | 51.6 |
| ImageBind | 2 | 44.2 | 44.7 | 55.4 | 50.9 | 52.5 | 50.5 | 48.6 | 61.8 | 53.0 |
| LanguageBind | 8 | 43.4 | 41.5 | 53.4 | 55.0 | 46.6 | 51.0 | 65.9 | 52.8 |
| *Video Multimodal Generative Models : Text + Multi-Frames as Input* | | | | | | | | | | |
| GPT-4o | 16 | 69.8 | 64.7 | **80.6** | 78.4 | **67.9** | 67.2 | 75.8 | 85.6 | 75.7 |
| Gemini-1.5-Pro | 1FPS | 67.4 | 60.1 | 70.6 | 70.7 | 58.7 | 59.5 | 67.9 | 79.2 | 67.5 |
| Claude-3.5-Sonnet | 8 | 62.0 | 57.4 | 70.7 | 70.3 | 60.0 | 57.8 | 61.3 | 76.2 | 65.5 |
| Qwen2-VL-72B | 32 | 72.1 | 69.2 | 79.9 | **78.7** | 65.9 | 69.5 | **76.0** | **85.7** | **75.8** |
| Qwen2-VL-7B | 32 | 65.9 | 45.8 | 67.3 | 66.1 | 54.6 | 54.7 | 69.7 | 75.7 | 64.4 |
| LLaVA-OneVision-72B | 8 | **73.6** | 56.0 | 76.2 | 70.3 | 65.2 | 62.4 | 73.2 | 84.2 | 72.1 |
| LLaVA-OneVision-7B | 32 | 63.6 | 45.5 | 62.9 | 56.9 | 52.8 | 54.0 | 66.5 | 77.1 | 61.9 |
| LLaVA-NeXT-Video-34B | 32 | 61.2 | 56.0 | 66.4 | 61.6 | 58.5 | 59.3 | 63.4 | 74.1 | 64.0 |
| LLaVA-NeXT-Video-7B | 8 | 69.0 | 65.7 | 68.2 | 62.2 | 66.5 | 68.6 | 52.2 | 74.3 | 65.1 |
| InternLM-XC2.5 | 1FPS | 55.8 | 42.5 | 62.7 | 62.5 | 52.6 | 51.1 | 58.3 | 70.7 | 58.8 |
| VideoLLaVA | 8 | 69.8 | **70.2** | 71.4 | 70.0 | 70.6 | **70.2** | 50.5 | 75.5 | 67.1 |
| MiniCPM-V2.6 | 1FPS | 59.4 | 52.3 | 65.5 | 62.5 | 54.1 | 53.3 | 63.5 | 74.7 | 62.3 |
| Phi-3.5-Vision | 2 | 53.5 | 55.3 | 60.1 | 55.9 | 54.0 | 52.2 | 55.3 | 69.4 | 58.0 |
| MA-LMM | 4 | 54.3 | 43.0 | 48.0 | 47.8 | 46.3 | 48.8 | 48.6 | 49.6 | 48.0 |
| $M^3$ | 6 | 51.9 | 53.6 | 58.9 | 56.3 | 52.2 | 53.7 | 50.8 | 68.6 | 56.4 |
| *Large Multimodal Models (LMMs): Text + 1 frame as Input* | | | | | | | | | | |
| GPT-4o | 1 | 67.4 | 65.1 | 74.1 | 70.3 | 64.2 | 62.6 | 68.7 | 78.4 | 70.0 |
| LLaVA-1.5-13B | 1 | 57.4 | 51.9 | 57.6 | 53.8 | 50.4 | 53.9 | 54.2 | 63.1 | 55.7 |
| LLaVA-1.5-7B | 1 | 62.0 | 61.5 | 62.2 | 54.1 | 61.4 | 64.9 | 51.0 | 67.9 | 60.5 |
| LLaVA-NeXT-34B | 1 | 51.2 | 55.7 | 61.2 | 60.0 | 54.8 | 53.0 | 65.0 | 67.5 | 60.5 |
| Phi-3-Vision | 1 | 46.5 | 45.5 | 56.0 | 55.6 | 48.8 | 49.2 | 56.9 | 62.1 | 54.4 |
| *Large Language Models (LLMs): Text as Input* | | | | | | | | | | |
| GPT-4o | 0 | 65.1 | 59.8 | 73.7 | 70.0 | 61.5 | 60.1 | 69.3 | 68.6 | 67.7 |
| Gemini-1.5-Pro | 0 | 54.3 | 42.5 | 60.4 | 62.2 | 53.6 | 53.3 | 64.8 | 57.4 | 58.1 |
| Yi-34B | 0 | 51.9 | 62.3 | 60.1 | 60.3 | 57.1 | 55.1 | 65.4 | 58.0 | 59.9 |
| Vicuna7b-1-5 | 0 | 55.8 | 47.2 | 51.7 | 48.4 | 50.1 | 49.4 | 49.9 | 51.4 | 50.5 |
| Flan-T5-XL | 0 | 53.5 | 57.7 | 60.2 | 59.7 | 56.1 | 56.9 | 54.9 | 60.7 | 57.9 |
| Flan-T5-XXL | 0 | 55.8 | 62.5 | 59.0 | 58.4 | 54.2 | 48.2 | 49.3 | 58.9 | 55.1 |

# E    PER SUBSET RESULTS FOR SHORT AND LONG VIDEO QA

The per subset results (denoted as "T-") for short and long video QA under Binary Accuracy (BA) are shown in Table 11, and Table 13, respectively. We also provide a visualization of per subset short video QA performance in Figure 10 (a). The per subset results (denoted as "T-") for long video QA under Multiple Binary Accuracy (MBA) is shown in Table 12. Still, human achieves much better performance than all multimodal videos. Interestingly, both human and AI models show worse performance on Finegym, the professional video understanding dataset.

# F    MORE RESULTS WITH EXTENDED FRAMES

In the main paper, we only report the performance of each multimodal video models with the the number of frames that leads to the best performance. Here we extend the results to show the results of more frames in Table 9 and Table 10. Note that we include results for models including LLaVA-Video Zhang et al. (2024c), Aria Li et al. (2024b) and LongVU Shen et al. (2024). Generally, with more frames, performance saturates around 8-16 frames, meaning that models struggle to improve fine-grained activity understanding even with more frames. This is a clear contrast with human performance, showing that there is a large space for multimodal video models to improve.

# G    DATA ANNOTATION PIPELINE AND PLATFORM

Our **six-stage** annotation process includes: (1) AMT workers filtering through an exam, (2) Initial positive captions by filtered AMT workers, (3) author (experts) refined positive captions, (4) LLM-generated negatives with targeted temporal changes, (5) meticulous author filtering, (6) AMT

Table 9: *TemporalBench* performance of various models under binary QA accuracy (BA) and multiple binary QA accuracy (MBA) setting for short and long question answering with different number of frames. "Overall" denotes the average performance of short and long video QA performance. (Part 1)

| Model | # Frames | Overall MBA | Overall BA | Short MBA | Short BA | Long MBA | Long BA | Captioning |
|---|---|---|---|---|---|---|---|---|
| Human Performance | - | - | - | **67.9** | **89.7** | - | - | - |
| Random Chance | - | 9.5 | 50.0 | 9.5 | 50.0 | 9.5 | 50.0 | - |
| XCLIP | 8 | 12.0 | 51.7 | 12.9 | 51.6 | 11.1 | 51.7 | - |
| ImageBind | 2 | 12.4 | 52.0 | 14.0 | 53.0 | 10.7 | 51.0 | - |
| LanguageBind | 8 | 13.3 | 52.2 | 14.5 | 52.8 | 12.0 | 51.6 | - |
| GPT-4o | 64 | **35.4** | **73.3** | 38.0 | **76.0** | **32.7** | **70.5** | **63.5** |
| | 32 | 32.9 | 71.5 | 38.3 | 75.9 | 27.4 | 67.0 | 63.2 |
| | 16 | 34.3 | 72.8 | **38.5** | 75.7 | 30.1 | 69.8 | 61.3 |
| | 8 | 32.9 | 72.0 | 37.4 | 75.1 | 28.3 | 68.8 | 60.3 |
| | 4 | 31.9 | 71.2 | 35.8 | 74.4 | 28.0 | 68.0 | 58.8 |
| | 2 | 30.3 | 70.3 | 33.3 | 72.7 | 27.3 | 67.8 | 55.3 |
| | 1 | 26.5 | 67.4 | 28.4 | 70.0 | 24.5 | 64.7 | 52.3 |
| | 0 | 27.4 | 67.7 | 26.5 | 67.7 | 28.2 | 67.6 | - |
| Gemini-1.5-Pro | 1FPS | 25.7 | 66.4 | 26.6 | 67.5 | 24.7 | 65.2 | 56.5 |
| | 0 | 18.7 | 60.2 | 16.1 | 58.1 | 21.2 | 62.2 | - |
| Claude-3.5-Sonnet | 16 | 23.2 | 64.2 | 23.5 | 65.9 | 22.9 | 62.4 | 54.1 |
| | 8 | 24.1 | 65.1 | 23.6 | 65.5 | 24.5 | 64.6 | 53.1 |
| | 4 | 23.2 | 64.2 | 23.1 | 64.8 | 23.3 | 63.6 | 51.9 |
| | 2 | 21.1 | 62.2 | 21.2 | 61.9 | 20.9 | 62.4 | 48.2 |
| | 1 | 18.7 | 58.9 | 18.4 | 58.5 | 18.9 | 59.3 | 41.0 |
| Qwen2-VL-72B | 32 | 31.7 | 70.2 | 38.3 | 75.8 | 25.0 | 64.5 | 56.1 |
| | 16 | 31.5 | 70.1 | 36.9 | 74.6 | 26.1 | 65.5 | 54.1 |
| | 8 | 30.1 | 68.9 | 34.0 | 73.1 | 26.2 | 64.7 | 51.4 |
| | 4 | 28.6 | 68.4 | 31.2 | 71.5 | 26.0 | 65.3 | 48.3 |
| | 2 | 27.3 | 67.6 | 27.5 | 69.2 | 27.1 | 66.0 | 43.9 |
| Qwen2-VL-7B | 32 | 21.8 | 62.1 | 24.7 | 64.4 | 18.8 | 59.7 | 51.9 |
| | 16 | 21.2 | 61.5 | 23.6 | 63.3 | 18.7 | 59.7 | 50.3 |
| | 8 | 19.2 | 59.4 | 21.1 | 61.1 | 17.2 | 57.7 | 48.4 |
| | 4 | 17.4 | 58.5 | 19.3 | 59.5 | 15.4 | 57.5 | 46.1 |
| | 2 | 16.4 | 56.9 | 17.7 | 57.8 | 15.0 | 56.0 | 42.0 |
| LLaVA-OneVision-72B | 32 | 26.6 | 66.6 | 30.7 | 70.5 | 22.4 | 62.7 | 53.9 |
| | 16 | 27.2 | 67.3 | 32.1 | 71.2 | 22.3 | 63.4 | 54.2 |
| | 8 | 28.1 | 67.9 | 33.0 | 72.1 | 23.1 | 63.6 | 55.0 |
| | 4 | 27.6 | 67.3 | 31.4 | 71.2 | 23.8 | 63.4 | 54.2 |
| | 2 | 25.7 | 66.3 | 29.2 | 69.6 | 22.1 | 63.0 | 51.1 |
| | 1 | 23.3 | 64.1 | 27.1 | 67.9 | 19.5 | 60.2 | 48.6 |
| LLaVA-OneVision-7B | 32 | 18.7 | 59.4 | 21.2 | 61.9 | 16.2 | 56.9 | 50.1 |
| | 16 | 17.9 | 58.8 | 20.1 | 60.9 | 15.6 | 56.6 | 50.4 |
| | 8 | 17.3 | 57.8 | 19.5 | 59.9 | 15.0 | 55.7 | 50.2 |
| | 4 | 16.4 | 56.3 | 18.9 | 58.9 | 13.9 | 53.7 | 49.7 |
| | 2 | 14.8 | 54.4 | 16.8 | 56.1 | 12.7 | 52.7 | 47.1 |
| | 1 | 12.0 | 51.4 | 13.4 | 53.3 | 10.6 | 49.5 | 44.1 |
| LLaVA-NeXT-Video-34B | 32 | 19.9 | 61.1 | 22.0 | 64.0 | 17.7 | 58.2 | 53.1 |
| | 16 | 20.3 | 61.1 | 21.8 | 63.7 | 18.7 | 58.4 | 53.3 |
| | 8 | 20.5 | 61.9 | 21.4 | 63.3 | 19.5 | 60.4 | 53.4 |
| | 4 | 20.4 | 61.7 | 20.7 | 63.0 | 20.0 | 60.3 | 52.5 |
| | 2 | 19.9 | 61.2 | 20.0 | 61.8 | 19.7 | 60.6 | 48.9 |
| | 1 | 19.0 | 59.8 | 19.0 | 60.5 | 18.9 | 59.1 | 46.2 |

worker/expert performance as validation. This ensures high-quality, challenging negatives. Specifically,

**Positive Captions**    We use Amazon Mechanical Turk (AMT) with in-context examples for positive caption annotation, shown in Figure 13. Then we use Label Studio to let authors refine the caption. As shown in Figure 14, authors can edit the caption from AMT workers. Also, we provide the original short video captions to let people better understand our task.

**Negative Captions Filtering Platform**    We first prompt LLMs (GPT-4o, Gemini, and Llama-3.1-405b) to get initial negative captions, and then ask authors to choose the negatives that can reflect the temporal dynamic. The visualization of the multi-choice platform in shown in Figure 15.

Table 10: *TemporalBench* performance of various models under binary QA accuracy (BA) and multiple binary QA accuracy (MBA) setting for short and long question answering with different number of frames. "Overall" denotes the average performance of short and long video QA performance. (Part 2)

| Model | # Frames | Overall MBA | Overall BA | Short MBA | Short BA | Long MBA | Long BA | Captioning |
|---|---|---|---|---|---|---|---|---|
| LLaVA-NeXT-Video-7B | 32 | 15.9 | 57.1 | 17.3 | 59.5 | 14.5 | 54.7 | 51.6 |
| | 16 | 19.3 | 59.9 | 22.4 | 64.0 | 16.1 | 55.7 | 49.9 |
| | 8 | 20.5 | 61.2 | 23.6 | 65.1 | 17.3 | 57.2 | 50.1 |
| | 4 | 20.0 | 60.7 | 23.0 | 64.2 | 17.0 | 57.2 | 49.2 |
| | 2 | 19.2 | 60.3 | 21.5 | 63.1 | 16.8 | 57.4 | 46.8 |
| | 1 | 17.6 | 59.1 | 19.1 | 62.0 | 16.1 | 56.1 | 44.0 |
| LLaVA-Video-72b | 128 | 33.3 | 72.1 | 36.3 | 74.9 | 30.2 | 69.4 | 53.7 |
| | 64 | 33.8 | 72.1 | 37.1 | 75.3 | 30.5 | 68.9 | 54.2 |
| | 32 | 33.7 | 72.4 | 37.7 | 75.9 | 29.6 | 68.8 | 54.8 |
| | 16 | 33.1 | 72.0 | 37.4 | 75.8 | 28.8 | 68.3 | 54.9 |
| | 8 | 32.7 | 71.9 | 36.3 | 75.6 | 29.0 | 68.3 | 54.7 |
| | 4 | 32.5 | 71.4 | 35.9 | 74.9 | 29.2 | 68.0 | 53.1 |
| | 2 | 31.3 | 70.4 | 33.0 | 73.4 | 29.7 | 67.5 | 50.2 |
| | 1 | 29.5 | 69.5 | 30.9 | 71.6 | 28.1 | 67.4 | 47.0 |
| LLaVA-Video-7b | 128 | 21.9 | 62.1 | 20.2 | 61.2 | 23.5 | 63.0 | 51.8 |
| | 64 | 22.8 | 63.0 | 22.0 | 62.7 | 23.6 | 63.3 | 52.5 |
| | 32 | 22.9 | 63.6 | 22.9 | 63.3 | 22.9 | 63.9 | 52.1 |
| | 16 | 21.6 | 63.0 | 21.8 | 62.9 | 21.5 | 63.1 | 51.9 |
| | 8 | 20.9 | 62.0 | 21.5 | 61.8 | 20.3 | 62.2 | 50.9 |
| | 4 | 18.8 | 60.5 | 18.4 | 59.8 | 19.1 | 61.2 | 49.6 |
| | 2 | 18.2 | 59.6 | 18.9 | 58.6 | 17.5 | 60.7 | 46.6 |
| | 1 | 16.0 | 57.0 | 16.2 | 56.7 | 15.9 | 57.4 | 43.1 |
| Aria | 128 | 25.1 | 66.0 | 27.2 | 68.7 | 23.1 | 63.4 | 50.6 |
| | 64 | 24.2 | 65.9 | 26.3 | 68.4 | 22.0 | 63.4 | 51.1 |
| | 32 | 25.0 | 65.9 | 26.6 | 68.4 | 23.4 | 63.5 | 51.5 |
| | 16 | 24.4 | 65.6 | 26.2 | 68.3 | 22.6 | 62.9 | 51.4 |
| | 8 | 24.4 | 65.4 | 26.5 | 68.5 | 22.3 | 62.3 | 51.7 |
| | 4 | 23.5 | 64.8 | 26.0 | 67.8 | 21.0 | 61.7 | 50.8 |
| | 2 | 22.3 | 62.2 | 25.1 | 66.5 | 19.6 | 57.8 | 48.7 |
| | 1 | 17.9 | 54.8 | 22.2 | 64.5 | 13.6 | 45.1 | 45.7 |
| LongVU | 1FPS | 18.9 | 58.5 | 20.9 | 61.7 | 16.9 | 55.3 | 40.5 |
| InternLM-XC2.5 | 1FPS | 16.8 | 57.3 | 17.9 | 58.8 | 15.6 | 55.8 | 52.4 |
| VideoLLaVA | 8 | 20.3 | 61.6 | 25.5 | 67.1 | 15.1 | 56.0 | 46.0 |
| MiniCPM-V2.6 | 1FPS | 20.4 | 61.3 | 21.4 | 62.3 | 19.3 | 60.3 | 47.2 |
| Phi-3.5-Vision | 32 | 14.1 | 54.3 | 15.6 | 56.8 | 12.6 | 51.7 | 48.4 |
| | 16 | 14.7 | 55.1 | 15.9 | 57.2 | 13.5 | 53.0 | 48.9 |
| | 8 | 14.9 | 55.6 | 15.9 | 57.4 | 13.8 | 53.7 | 48.3 |
| | 4 | 15.0 | 56.0 | 15.5 | 57.5 | 14.5 | 54.5 | 44.0 |
| | 2 | 15.5 | 56.2 | 16.9 | 58.0 | 14.1 | 54.4 | 42.9 |
| | 1 | 14.5 | 55.9 | 16.5 | 57.7 | 12.5 | 54.0 | 42.1 |
| MA-LMM | 4 | 9.1 | 47.4 | 9.2 | 48.0 | 9.0 | 46.9 | 38.7 |
| M3 | 6 | 13.3 | 54.7 | 14.8 | 56.4 | 11.8 | 53.1 | 47.8 |
| LLaVA-1.5-13B | 1 | 13.7 | 55.1 | 13.1 | 55.7 | 14.2 | 54.5 | 47.9 |
| LLaVA-1.5-7B | 1 | 15.3 | 56.8 | 18.3 | 60.5 | 12.3 | 53.2 | 45.7 |
| LLaVA-NeXT-34B | 1 | 19.0 | 60.5 | 18.0 | 60.5 | 19.9 | 60.5 | 49.1 |
| Phi-3-Vision | 1 | 15.4 | 55.2 | 15.1 | 54.4 | 15.6 | 56.0 | 42.0 |
| Gemini-1.5-Pro | 0 | 18.6 | 60.1 | 16.1 | 58.1 | 21.2 | 62.2 | - |
| Yi-34B | 0 | 18.5 | 59.7 | 18.7 | 59.9 | 18.4 | 59.5 | - |
| Vicuna7b-1-5 | 0 | 10.1 | 50.8 | 10.4 | 50.5 | 9.9 | 51.1 | - |
| Flan-T5-XL | 0 | 18.6 | 59.0 | 17.9 | 57.9 | 19.4 | 60.1 | - |
| Flan-T5-XXL | 0 | 15.9 | 56.0 | 15.1 | 55.1 | 16.7 | 56.9 | - |

## H   VIDEO STATISTICS

Video length distribution of (a) short video clips and (b) long videos in *TemporalBench* is shown in Figure 16. For benchmarks without caption, we computed the number of words per second using the length of question and answers.

For ∼10K short video QA, Qwen2VL-7B (16 frames) requires 34 minutes for MBA evaluation on 4 A6000 GPUs, comparable to existing benchmarks (e.g., 32 minutes for NextQA Xiao et al. (2021) which owns multichoice QA format).

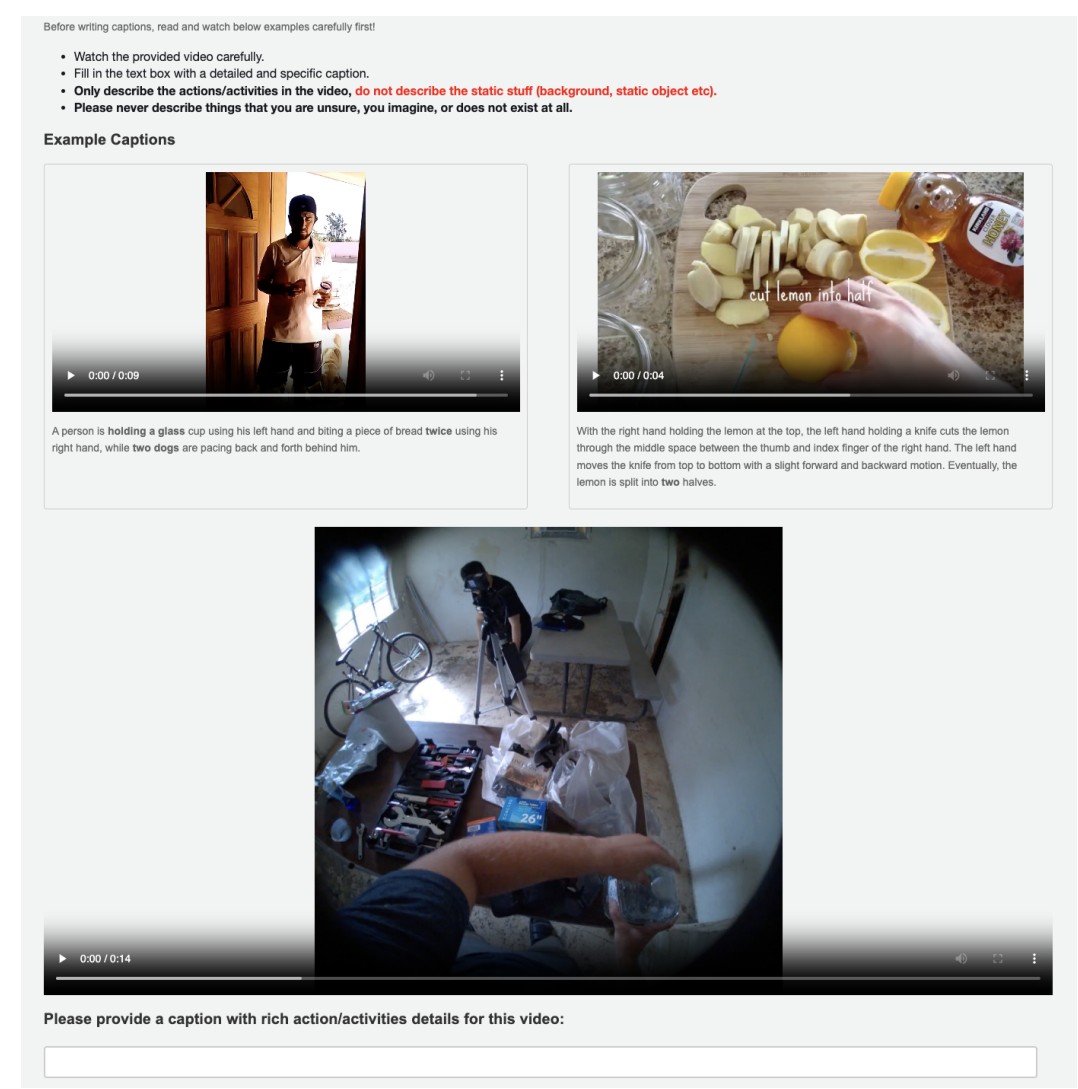

Figure 13: Positive caption annotation platform for AMT workers.

In *TemporalBench*, each sub-question is answered with high human accuracy (*e.g.*, , ~92% BA on most categories), but the *product* of probabilities will lead to an overall lower success rate ($92\%^5 = 65.9\%$). SOTA models (GPT-4o) still lag significantly (38.5% on short video MBA), highlighting *TemporalBench*'s difficulty.

## I    MORE RELATED WORKS

Existing works explored video understanding along different directions. YouCook2 Zhou et al. (2018) emphasizes procedure videos but lacks fine-grained temporal understanding. VITATECS Li et al. (2024d), TempCompass Liu et al. (2024c), TOMATO Shangguan et al. (2024) and TVBench Cores et al. (2024) work towards better model temporal dynamics via the counterfactual manner, but still lacks the dense captions for fine-grained details.

## J    PROMPTS FOR NEGATIVE CAPTION CURATION

We provide the prompts for the negative caption curation in Table 14 and Table 15 for the word level replacement and event level re-ordering for the negative caption curation process, respectively.

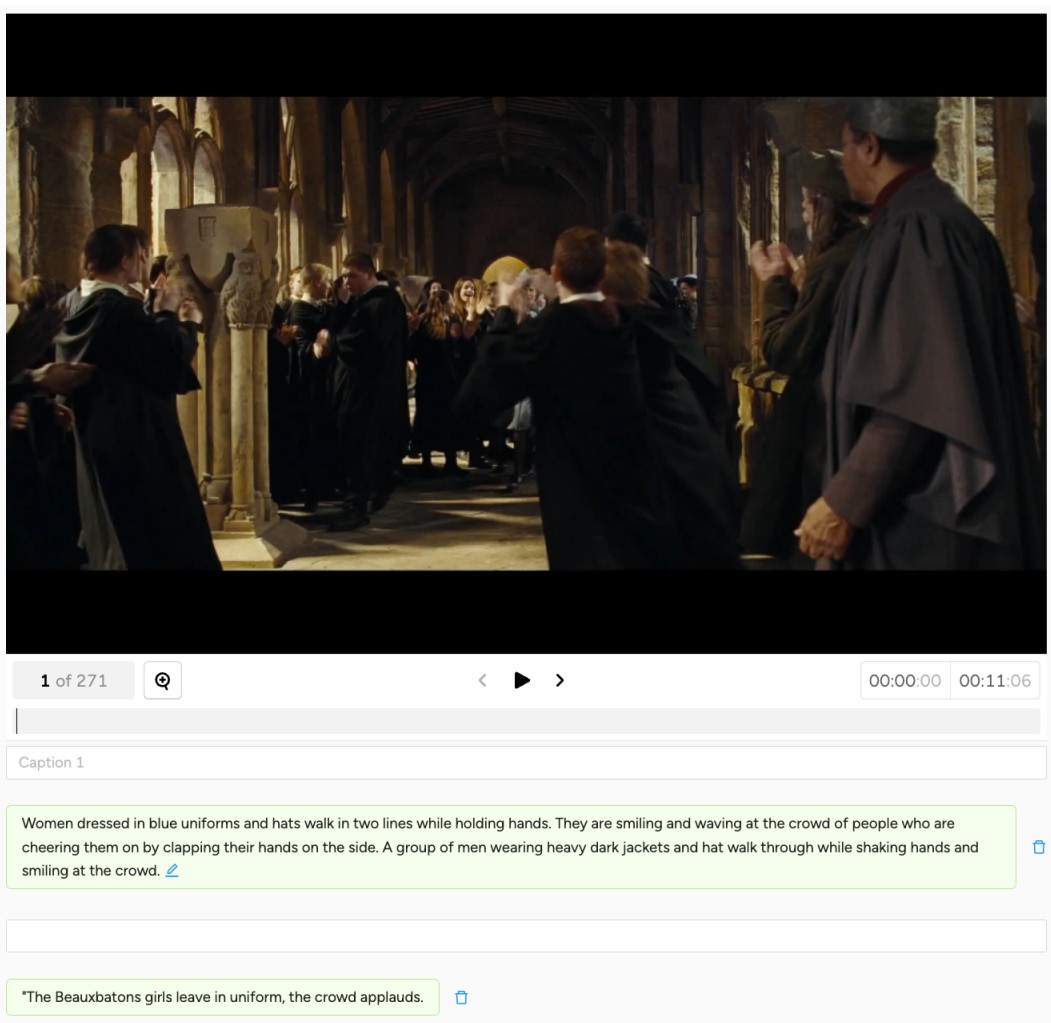

Figure 14: Positive caption refinement platform.

We leverage such prompts to gather responses from LLMs including including GPT-4o, Llama-3.1-405B, and Gemini-1.5-Pro, followed by human filtering.

## K  LIMITATIONS

One cannot fully analyze the behavior of proprietary models included in this paper due to the lack of access to these models, which are GPT-4o, Gemini-1.5-Pro and Claude 3.5 Sonnet.

## L  BROADER IMPACT

*TemporalBench*, a comprehensive benchmark for video understanding, has the potential to significantly advance research in this field by offering improved metrics for model evaluation. Our work aims to enhance the temporal reasoning capabilities of future video understanding models. However, the broader impact of more advanced video understanding technologies raises important societal concerns, including the risk of mass surveillance, privacy violations, and the development of harmful applications like autonomous weapons. Therefore, we strongly encourage thoughtful consideration when deploying these models in real-world scenarios to mitigate negative or unintended consequences.

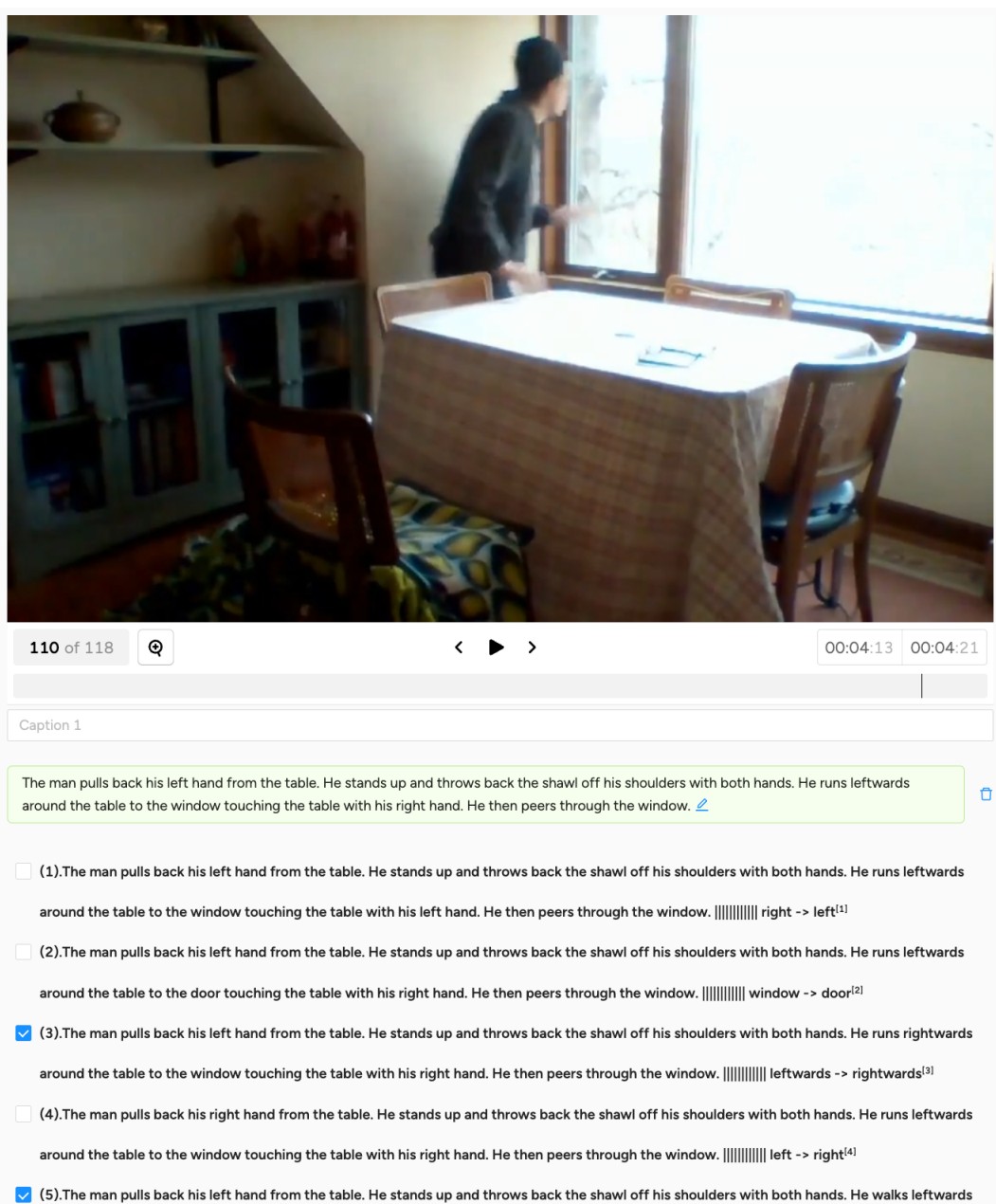

Figure 15: Negative caption annotation platform.

## M    THE USE OF LLMs IN THIS RESEARCH

LLMs including Gemini and Claude are to help polish this paper, LLMs are also used during the negative caption generation process.

## N    TOWARDS IMPROVING TEMPORAL FINE-GRAINED CAPABILITY

Here we explore enhancing video LLM's temporal fine-grained capability, where we focus on curating high-quality temporal negatives. By leveraging LLaVA-Video's dense captions and Gemini 2.5 Pro, we curate 5k high quality captioning binary QA problems. Finetuning upon Qwen-2.5-VL-

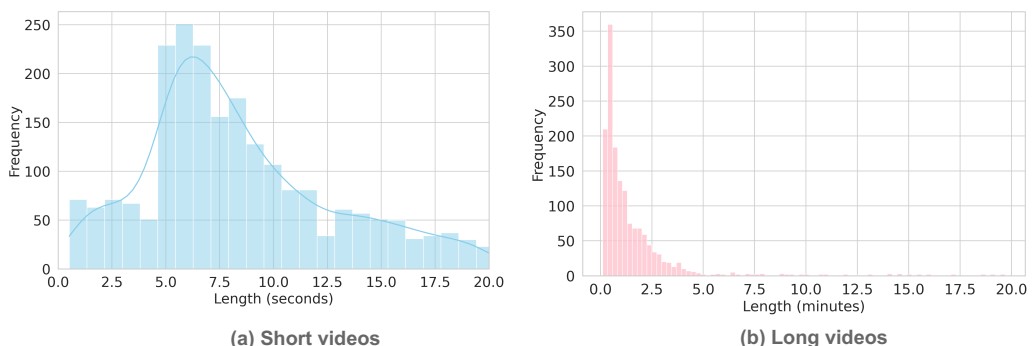

Figure 16: Video length distribution of (a) short video clips and (b) long videos in *TemporalBench*.

Instruct-7B leads to 3.5% improvements on the MBA score, which shows the positive signal towards our goal.

Table 11: *TemporalBench* performance of various multimodal generative models and embedding models under the binary QA accuracy (BA) and multiple binary QA settings (MBA) for **short videos**. The prefix "T-" indicates **BA** performance for the annotated subset in our *TemporalBench*. We show the result with the best average MBA performance for each model with respect to the number of frames, denoted as # Frames.

| Model | # Frames | T-ActivityNet | T-Charades | T-FineGym | T-Movie | T-Oops | T-COIN | T-EgoExo4D | BA | MBA |
|---|---|---|---|---|---|---|---|---|---|---|
| Human Performance | - | **91.1** | **93.8** | **77.0** | **93.1** | **92.6** | **90.2** | **92.5** | **89.7** | **67.9** |
| Random Chance | - | 50.0 | 50.0 | 50.0 | 50.0 | 50.0 | 50.0 | 50.0 | 50.0 | 50.0 |
| **Video Embedding Models: Text + Multiple Frames as Input** | | | | | | | | | | |
| XCLIP | 8 | 52.7 | 52.8 | 49.0 | 53.9 | 53.5 | 52.3 | 48.1 | 51.6 | 12.9 |
| ImageBind | 2 | 52.9 | 52.6 | 47.5 | 55.4 | 56.8 | 52.4 | 53.4 | 53.0 | 14.0 |
| LanguageBind | 8 | 56.5 | 50.1 | 48.2 | 55.8 | 55.1 | 51.1 | 52.8 | 52.8 | 14.5 |
| **Video Multimodal Generative Models : Text + Multiple Frames as Input** | | | | | | | | | | |
| GPT-4o | 16 | **78.5** | **74.8** | 64.8 | 77.2 | **77.9** | **79.2** | 78.3 | 75.7 | 38.5 |
| Gemini-1.5-Pro | 1FPS | 70.7 | 63.0 | 55.0 | 72.5 | 70.3 | 70.2 | 70.8 | 67.5 | 26.6 |
| Claude-3.5-Sonnet | 8 | 68.5 | 62.4 | 62.7 | 68.2 | 64.2 | 65.4 | 66.8 | 65.5 | 23.6 |
| Qwen2-VL-72B | 32 | 76.6 | 74.5 | 65.4 | **79.8** | 77.7 | 77.2 | **79.7** | 75.8 | 38.3 |
| Qwen2-VL-7B | 32 | 67.0 | 65.2 | 49.9 | 70.5 | 66.5 | 66.5 | 66.6 | 64.4 | 24.7 |
| LLaVA-OneVision-72B | 8 | 76.0 | 70.4 | 59.3 | 76.1 | 75.2 | 73.5 | 74.9 | 72.1 | 33.0 |
| LLaVA-OneVision-7B | 32 | 66.5 | 60.0 | 49.4 | 68.0 | 61.6 | 64.6 | 64.4 | 61.9 | 21.2 |
| LLaVA-NeXT-Video-34B | 32 | 67.5 | 62.9 | 56.3 | 68.0 | 66.1 | 63.4 | 64.5 | 64.0 | 22.0 |
| LLaVA-NeXT-Video-7B | 8 | 68.0 | 66.5 | 56.7 | 69.9 | 66.1 | 65.2 | 65.0 | 65.1 | 23.6 |
| InternLM-XC2.5 | 1FPS | 61.0 | 57.9 | 50.6 | 63.5 | 60.3 | 59.2 | 59.7 | 58.8 | 17.9 |
| VideoLLaVA | 8 | 71.8 | 63.4 | 61.6 | 68.2 | 68.5 | 68.9 | 67.3 | 67.1 | 25.5 |
| MiniCPM-V2.6 | 1FPS | 66.1 | 59.6 | 54.1 | 68.0 | 63.1 | 62.7 | 62.7 | 62.3 | 21.4 |
| Phi-3.5-Vision | 2 | 62.0 | 55.8 | 50.0 | 64.1 | 58.2 | 57.7 | 58.9 | 58.0 | 16.9 |
| MA-LMM | 4 | 49.8 | 48.8 | 42.3 | 48.0 | 49.9 | 49.0 | 48.8 | 48.0 | 9.4 |
| $M^3$ | 6 | 59.5 | 54.9 | 51.1 | 60.9 | 58.9 | 54.9 | 55.2 | 56.4 | 14.8 |
| **Large Multimodal Models (LMMs): Text + 1 Frame as Input** | | | | | | | | | | |
| GPT-4o | 1 | 69.1 | 67.1 | 64.8 | 71.0 | 71.9 | 71.0 | 74.0 | 70.0 | 28.4 |
| LLaVA-1.5-13B | 1 | 57.6 | 54.3 | 51.9 | 56.8 | 53.2 | 58.1 | 57.8 | 55.7 | 13.1 |
| LLaVA-1.5-7B | 1 | 64.2 | 58.6 | 55.7 | 61.0 | 57.5 | 62.7 | 63.9 | 60.5 | 18.3 |
| LLaVA-NeXT-34B | 1 | 59.7 | 60.3 | 55.0 | 61.8 | 62.0 | 61.0 | 63.7 | 60.5 | 18.0 |
| Phi-3-Vision | 1 | 57.4 | 54.5 | 45.2 | 57.5 | 52.8 | 55.8 | 58.9 | 54.4 | 15.1 |
| **Large Language Models (LLMs): Text as Input** | | | | | | | | | | |
| GPT-4o | 0 | 66.2 | 67.4 | **65.6** | 65.6 | 68.9 | 67.8 | 71.7 | 67.7 | 26.5 |
| Gemini-1.5-Pro | 0 | 58.5 | 57.6 | 50.6 | 59.8 | 57.6 | 58.6 | 64.3 | 58.1 | 16.1 |
| Yi-34B | 0 | 59.1 | 62.3 | 54.9 | 59.7 | 57.7 | 63.1 | 63.6 | 59.9 | 18.7 |
| Vicuna7b-1-5 | 0 | 49.7 | 49.5 | 50.2 | 50.7 | 50.5 | 50.0 | 52.1 | 50.5 | 10.4 |
| Flan-T5-XL | 0 | 60.5 | 59.2 | 50.5 | 60.7 | 56.8 | 58.7 | 60.3 | 57.9 | 17.9 |
| Flan-T5-XXL | 0 | 56.7 | 49.3 | 52.0 | 59.0 | 54.6 | 56.1 | 56.2 | 55.1 | 15.1 |

Table 12: *TemporalBench* performance of various multimodal generative models and embedding models under **long video** understanding with binary QA accuracy (BA) and multiple binary QA accuracy (MBA). The **MBA** performance under each dataset is also included. We show the result with the best average MBA performance for each model with respect to the number of frames, denoted as # Frames.

| Model | # Frames | T-ActivityNet | T-Charades | T-FineGym | T-COIN | T-EgoExo4D | BA | MBA |
|---|---|---|---|---|---|---|---|---|
| Random Performance | - | 9.3 | 9.8 | 10.1 | 11.4 | 9.3 | 50.0 | 9.5 |
| **Video Embedding Models: Text + Multi-Frames as Input** | | | | | | | | |
| XCLIP | 8 | 11.1 | 12.4 | 6.5 | 10.8 | 11.8 | 51.7 | 11.1 |
| ImageBind | 2 | 10.2 | 8.1 | 9.3 | 10.8 | 12.4 | 51.0 | 10.7 |
| LanguageBind | 8 | 11.7 | 10.8 | 10.3 | 11.0 | 14.1 | 51.6 | 12.0 |
| **Video Multimodal Generative Models : Text + Multi-Frames as Input** | | | | | | | | |
| GPT-4o | 64 | **40.0** | **37.8** | 16.8 | **32.7** | 29.3 | **70.5** | **32.7** |
| Gemini-1.5-Pro | 1FPS | 32.1 | 18.4 | 18.7 | 24.8 | 23.8 | 65.2 | 24.7 |
| Claude-3.5-Sonnet | 8 | 28.9 | 22.2 | 16.8 | 22.2 | 26.7 | 64.6 | 24.5 |
| Qwen2-VL-72B | 8 | 32.4 | 20.5 | 21.5 | 18.9 | 33.1 | 64.7 | 26.2 |
| Qwen2-VL-7B | 32 | 22.2 | 20.0 | 9.3 | 18.3 | 18.7 | 59.7 | 18.8 |
| LLaVA-OneVision-72B | 4 | 28.6 | 19.5 | 18.7 | 16.5 | 30.9 | 63.4 | 23.8 |
| LLaVA-OneVision-7B | 32 | 21.3 | 13.0 | 13.1 | 11.4 | 19.8 | 56.9 | 16.2 |
| LLaVA-NeXT-Video-34B | 4 | 23.5 | 22.2 | 19.6 | 17.9 | 19.2 | 60.3 | 20.0 |
| LLaVA-NeXT-Video-7B | 8 | 18.1 | 21.6 | 10.3 | 18.5 | 15.6 | 57.2 | 17.3 |
| InternLM-XC2.5 | 1FPS | 21.0 | 18.4 | 20.6 | 14.0 | 11.4 | 55.8 | 15.6 |
| VideoLLaVA | 8 | 20.0 | 16.8 | 15.9 | 9.8 | 16.6 | 56.0 | 15.1 |
| MiniCPM-V2.6 | 1FPS | 14.3 | 16.8 | 6.5 | 17.1 | 14.1 | 60.3 | 19.3 |
| Phi-3.5-Vision | 4 | 23.2 | 11.9 | 19.6 | 10.2 | 13.3 | 54.5 | 14.5 |
| MA-LMM | 4 | 10.2 | 9.2 | 2.8 | 11.4 | 11.6 | 47.1 | 9.2 |
| $M^3$ | 6 | 10.8 | 8.6 | 12.1 | 13.0 | 12.4 | 53.1 | 11.8 |
| **Large Multimodal Models (LMMs): Text + 1 frame as Input** | | | | | | | | |
| GPT-4o | 1 | 27.9 | 23.2 | 19.6 | 25.2 | 22.9 | 64.7 | 24.5 |
| LLaVA-1.5-13B | 1 | 14.3 | 11.9 | 10.3 | 15.4 | 14.7 | 54.8 | 14.2 |
| LLaVA-1.5-7B | 1 | 9.2 | 11.9 | 10.3 | 12.8 | 14.5 | 53.2 | 12.3 |
| LLaVA-NeXT-34B | 1 | 21.6 | 20.5 | 19.6 | 18.9 | 19.8 | 60.5 | 19.9 |
| Phi-3-Vision | 1 | 18.1 | 12.4 | 15.0 | 15.4 | 15.6 | 56.0 | 15.6 |
| **Large Language Models (LLMs): Text as Input** | | | | | | | | |
| GPT-4o | 0 | 27.6 | 32.4 | 17.8 | 24.2 | **33.5** | 67.6 | 28.2 |
| Gemini-1.5-Pro | 0 | 22.9 | 19.5 | 17.8 | 19.3 | 23.4 | 62.2 | 21.2 |
| Yi-34B | 0 | 19.7 | 19.5 | 14.0 | 15.9 | 20.6 | 59.5 | 18.4 |
| Vicuna7b-1-5 | 0 | 6.3 | 9.2 | 9.3 | 10.6 | 12.0 | 51.1 | 9.9 |
| Flan-T5-XL | 0 | 21.6 | 15.7 | **23.4** | 18.1 | 19.8 | 60.1 | 19.4 |
| Flan-T5-XXL | 0 | 20.0 | 11.9 | 18.7 | 15.7 | 17.1 | 56.9 | 16.7 |

Table 13: *TemporalBench* performance of various multimodal generative models and embedding models under **long video** understanding with binary QA accuracy (BA) and multiple binary QA accuracy (MBA). The **BA** performance under each dataset is also included. We show the result with the best average MBA performance for each model with respect to the number of frames, denoted as # Frames.

| Model | # Frames | T-ActivityNet | T-Charades | T-FineGym | T-COIN | T-EgoExo4D | BA | MBA |
|---|---|---|---|---|---|---|---|---|
| Random Performance | - | 50.0 | 50.0 | 50.0 | 50.0 | 50.0 | 50.0 | 50.0 |
| **Video Embedding Models: Text + Multi-Frames as Input** | | | | | | | | |
| XCLIP | 8 | 51.9 | 48.7 | 47.9 | 52.6 | 52.8 | 51.7 | 11.1 |
| ImageBind | 2 | 50.3 | 52.6 | 47.9 | 51.3 | 51.3 | 51.0 | 10.7 |
| LanguageBind | 8 | 51.9 | 46.4 | 48.2 | 52.0 | 53.7 | 51.6 | 12.0 |
| **Video Multimodal Generative Models : Text + Multi-Frames as Input** | | | | | | | | |
| GPT-4o | 64 | **74.8** | **73.8** | 61.2 | **70.1** | 68.7 | **70.5** | **32.7** |
| Gemini-1.5-Pro | 1FPS | 67.0 | 61.6 | 60.6 | 65.9 | 65.9 | 65.2 | 24.7 |
| Claude-3.5-Sonnet | 8 | 66.8 | 63.7 | 56.7 | 63.1 | 66.6 | 64.6 | 24.5 |
| Qwen2-VL-72B | 8 | 68.5 | 59.6 | 62.5 | 59.6 | 70.0 | 64.7 | 26.2 |
| Qwen2-VL-7B | 32 | 60.7 | 58.0 | 49.9 | 59.8 | 61.9 | 59.7 | 18.8 |
| LLaVA-OneVision-72B | 4 | 67.0 | 63.5 | 61.2 | 55.8 | 69.3 | 63.4 | 23.8 |
| LLaVA-OneVision-7B | 32 | 60.0 | 53.6 | 57.6 | 53.2 | 59.8 | 56.9 | 16.2 |
| LLaVA-NeXT-Video-34B | 4 | 59.4 | 63.0 | 57.6 | 59.5 | 61.4 | 60.3 | 20.0 |
| LLaVA-NeXT-Video-7B | 8 | 60.9 | 58.6 | 51.5 | 56.7 | 56.1 | 57.2 | 17.3 |
| InternLM-XC2.5 | 1FPS | 59.6 | 58.9 | 57.0 | 54.9 | 52.8 | 55.8 | 15.6 |
| VideoLLaVA | 8 | 61.2 | 57.0 | 59.5 | 50.1 | 57.3 | 56.0 | 15.1 |
| MiniCPM-V2.6 | 1FPS | 53.7 | 58.6 | 41.3 | 54.8 | 53.9 | 60.3 | 19.3 |
| Phi-3.5-Vision | 4 | 60.3 | 52.3 | 58.1 | 50.3 | 55.1 | 54.5 | 14.5 |
| MA-LMM | 4 | 47.4 | 51.7 | 36.4 | 50.1 | 51.2 | 47.1 | 9.2 |
| $M^3$ | 6 | 52.5 | 52.9 | 51.0 | 53.4 | 53.6 | 53.1 | 11.8 |
| **Large Multimodal Models (LMMs): Text + 1 frame as Input** | | | | | | | | |
| GPT-4o | 1 | 67.6 | 64.3 | 62.8 | 65.9 | 62.0 | 64.7 | 24.5 |
| LLaVA-1.5-13B | 1 | 55.1 | 52.3 | 52.9 | 55.0 | 54.8 | 54.5 | 14.2 |
| LLaVA-1.5-7B | 1 | 51.2 | 53.4 | 51.5 | 51.8 | 56.2 | 53.2 | 12.3 |
| LLaVA-NeXT-34B | 1 | 60.6 | 60.8 | 57.0 | 59.8 | 61.8 | 60.5 | 19.9 |
| Phi-3-Vision | 1 | 56.9 | 53.9 | 52.1 | 55.6 | 57.6 | 56.0 | 15.6 |
| **Large Larguage Models (LLMs): Text as Input** | | | | | | | | |
| GPT-4o | 0 | 67.1 | 68.1 | **63.6** | 65.1 | **71.3** | 67.6 | 28.2 |
| Gemini-1.5-Pro | 0 | 62.8 | 59.4 | 55.6 | 60.7 | 65.7 | 62.2 | 21.2 |
| Yi-34B | 0 | 59.0 | 60.2 | 56.5 | 59.5 | 60.4 | 59.5 | 18.4 |
| Vicuna7b-1-5 | 0 | 49.0 | 52.4 | 49.3 | 51.2 | 52.2 | 51.1 | 9.9 |
| Flan-T5-XL | 0 | 61.3 | 57.7 | 59.8 | 58.8 | 61.7 | 60.1 | 19.4 |
| Flan-T5-XXL | 0 | 59.4 | 53.6 | 59.5 | 56.3 | 56.5 | 56.9 | 16.7 |

Table 14: For each query, we illustrate the prompt construction process of the **word level replacement** for LLMs powered negative caption curation including GPT-4o, Llama-3.1-405B, and Gemini-1.5-Pro, to collect `query['response']` from `query['context']`, using few-shot in-context-learning, where examples are from `fewshot_samples`, each example including input `sample['context']` and output `sample['response']`. Note that `messages` is the final prompt. In this example, we provide the prompt used to generate the negative caption response.

```
messages = [ {"role":"system", "content": f""""You are a helpful assistant in generating negative captions for
videos designed to output JSON.
You are given a video caption. You are tasked to: Generate negative captions that changes the action descriptions being discussed
in exactly one of the entities by only changing or swapping one word or phrase.
For example, you can change "open the drawer" to "close the drawer", or change from "pull" to "push", or "quickly" to
"slowly". You should generate negatives that can not be solved by feeding one random frame into a multimodal model. For
example, changing "wood chair" into "plastic chair" is a bad practice. But changing "man playing with one cat" to "man playing
with two cats" is good. Swapping "left leg" to "right leg" is also good if this caption updates completely changes the motion in
the video.
All in all, the new description must meet all of these requirements:
1. The change of action descriptions must be sufficiently different to make the new description inaccurate, but it should also be
somewhat related to be challenging to an AI model.
2. Compared to the original description, the new description must differ in only one aspect. All other details must be kept the
same.
3. The new description must mimic the sentence structure of the original description.
4. The new description must be fluent, logical, and grammatically correct.
5. Pose challenging(difficult enough) negative captions so that a large multimodal text generation model should struggle to
distinguish the original caption v.s. negative captions.
Here are some examples whose output format you can have a reference:"""}]

fewshot_samples = [
{ "context": """ Original Caption: "A person is holding a glass cup using his left hand and biting a piece of bread twice
using his right hand, while two dogs are pacing back and forth behind him."
Your answer: """,
"response":"""[ "A person is holding a glass cup using his left hand and biting a piece of bread three times using his right
hand, while two dogs are pacing back and forth behind him.",
"A person is drinking water from a glass cup using his left hand and biting a piece of bread twice using his right hand, while
two dogs are pacing back and forth behind him.",
"A person is holding a glass cup using his left hand and biting a piece of bread twice using his right hand, while one dog is
pacing back and forth behind him.", ]""" },
{ "context": """ Original Caption: "With the right hand holding the lemon at the top, the left hand holding a knife cuts the
lemon through the middle space between the thumb and index finger of the right hand. The left hand moves the knife from top
to bottom with a slight forward and backward motion. Eventually, the lemon is split into two halves."
Your answer: """,
"response":"""[ "With the right hand holding the lemon at the top, the left hand holding a knife cuts the lemon through the
middle space between the thumb and index finger of the right hand. The left hand moves the knife from top to bottom with a
slight forward and backward motion. Eventually, the lemon is split into four pieces.",
"With the right hand holding the lemon at the top, the left hand holding a knife cuts the ginger through the middle space
between the thumb and index finger of the right hand. The left hand moves the knife from top to bottom with a slight forward
and backward motion. Eventually, the ginger becomes lots of pieces.",
"With the left hand holding the lemon at the top, the right hand holding a knife cuts the lemon through the middle space
between the thumb and index finger of the right hand. The right hand moves the knife from top to bottom with a slight forward
and backward motion. Eventually, the lemon is split into two halves.", ] """,
} ]

for sample in fewshot_samples:
    messages.append({"role":"user", "content":sample["context"]})
    messages.append({"role":"assistant", "content":sample['response']} )
messages.append({"role":"user", "content":'\n'.join(query)})
```

Table 15: For each query, we illustrate the prompt construction process of the **event level re-ordering** for LLMs powered negative caption curation including GPT-4o, Llama-3.1-405B, and Gemini-1.5-Pro, to collect `query['response']` from `query['context']`, using few-shot in-context-learning, where examples are from `fewshot_samples`, each example including input `sample['context']` and output `sample['response']`. Note that `messages` is the final prompt. In this example, we provide the prompt used to generate the negative caption response.

```
messages = [ { "role":"system", "content": f"""You are a helpful assistant tasked with generating negative cap-
tions for videos. Your goal is to output results in JSON format. You are provided with a video caption, and your job is to:
- Generate a "negative" version of the caption by rearranging the sequence of actions in the original caption. This rearrangement
should still make logical sense in real life.
Guidelines:
1. The changed order must still be reasonable. If reversing or rearranging the order of events is not logically possible, the
negative caption is not valid. For instance, if the original caption is "AAAA then BBBB" (e.g., a fire starting before people use
water to extinguish it), switching to "BBBB then AAAA" would be invalid.
2. The negative caption should differ only in the order of actions or events, while keeping the content of the events intact.
3. If there is only one event or all events happen at the same time (concurrently), output an empty array '[]'.
4. The negative caption must remain fluent, logical, and grammatically correct.
5. The negative captions should be challenging enough that a large multimodal model would find it difficult to distinguish
between the original and negative captions.
Here are some examples whose output format you can have a reference:"""}]

fewshot_samples = [
{ "context": """Original Caption: "A person is spraying cleaning liquid on the wooden chair in the external environment,
first from bottom to top on the right handrail, then top to bottom on one of the supporting pillars."
Your answer: """,
"response":"""[ "A person is spraying cleaning liquid on the wooden chair in the external environment, first from top to
bottom on one of the supporting pillars, then bottom to top on the right handrail", ]""" },
{ "context": """ Original Caption: "A person leans back pulling the rope as the boat sails forward. In the background more
boats sail on the sea. The waves in the sea ebbs and flows. There is a also a text "Accelerate" appearing at the bottom of the
scene."
Your answer: """,
"response":"""[] """,
},
{ "context": """ Original Caption: "A man uses his left hand to put the unscrewed cap onto the table, then uses the same
hand to move a red cup closer to his right hand. He is then seen pouring something from the right hand into the left cup. He
then moves the cup back onto the table and picks the cap back up to screw it back on."
Your answer: """,
"response":"""[ "A man uses his left hand to move a red cup closer to his right hand, then uses the same hand to put the
unscrewed cap onto the table. He is then seen pouring something from the right hand into the left cup. He then moves the cup
back onto the table and picks the cap back up to screw it back on.",
"A man is seen pouring something from the right hand into the left cup. Then he uses his left hand to put the unscrewed cap
onto the table, then uses the same hand to move a red cup closer to his right hand. He then moves the cup back onto the table
and picks the cap back up to screw it back on.",
"A man uses his left hand to put the unscrewed cap onto the table, then seen pouring something from the right hand into the left
cup. He then uses the left hand to move a red cup closer to his right hand. He then moves the cup back onto the table and picks
the cap back up to screw it back on.",
"A man uses his left hand to move a red cup closer to his right hand, then seen pouring something from the right hand into the
left cup. He then uses the left hand to put the unscrewed cap onto the table. He then moves the cup back onto the table and
picks the cap back up to screw it back on.",
"A man is seen pouring something from the right hand into the left cup. Then he uses the left hand to move a red cup closer to
his right hand, then uses his left hand to put the unscrewed cap onto the table. He then moves the cup back onto the table and
picks the cap back up to screw it back on.", ]""",
} ]

for sample in fewshot_samples:
    messages.append({"role":"user", "content":sample["context"]})
    messages.append({"role":"assistant", "content":sample['response']} )
messages.append({"role":"user", "content":'\n'.join(query)})
```