# OpenReview forum: "TemporalBench: Evaluating Fine-Grained Temporal Dynamics Understanding for Multimodal Models"
_ICLR.cc/2026/Conference — ICLR 2026 Conference Withdrawn Submission_

### Official Review · Reviewer_eL8f · 2025-11-01

**Soundness:** 2
**Presentation:** 2
**Contribution:** 2
**Rating:** 2
**Confidence:** 4

**Summary:**

This paper presents TemporalBench, a benchmark designed to evaluate fine-grained temporal reasoning in multimodal models. The benchmark contains ~15k video QAs derived from ~2k human annotations. The authors conduct experiments on video embedding models and video multimodal generative models to show performance gaps between humans and current models. To address biases arising from deriving incorrect multiple choice options from a “centered” correct one, the authors propose and analyze a new evaluation metric called Multiple Binary Accuracy (MBA).

**Strengths:**

1. The paper is clearly written and easy to follow.
2. The benchmark is well-motivated, and the annotation process is rigorous to ensure high quality annotations and QAs.

**Weaknesses:**

1. My primary concern is whether the benchmark’s design and experimental setup are consistent with the motivation and novelty stated by the authors.\
    (a). As the authors point out some previous video benchmarks resemble static image benchmarks. However, in the examples given in Figure 1, it appears that to distinguish “right to left” from “left to right”, and “push” from “pull” is still static image recognition but just needed to be done twice. Models are only required to identify “key frames” in order to correctly answer the questions, and thus the necessity of the video modality is not clearly enforced in this case. \
    (b). QA categories such as Action Frequency are not new and have already been introduced in several existing benchmarks (some before the LLM era).\
    (c). The claim that some more recent work “work towards better model temporal dynamics via the counterfactual manner, but still lack the dense captions for fine-grained details” is reasonable. However, it remains unclear how having such captions necessitates strong or novel temporal reasoning, given the concerns above.
2. Another concern lies with the necessity and significance of the proposed evaluation metric Multiple Binary Accuracy (MBA). As the authors mention that bias could be induced from deriving incorrect multiple choice options from a “centered” correct option, this claim is not supported by any comparison experiment on other MCQ-based benchmarks. On the contrary, some existing benchmarks adopt text-only Frequent Choice setting as a baseline to show that models are not biased by the phrasing of the options. Moreover, when the pairwise comparison is not between direct antonyms, one of the choices might still seem like the “centered” correct option and thus MBA would still be insufficient to eliminate such bias. (See more in Question 2)
3. It is unclear what the x-axis of Figure 3 is trying to demonstrate.
4. (Minor) Unfinished sentence in the last paragraph of the “Negative Caption Annotation” section.

**Questions:**

1. What message are the authors trying to deliver with the Figure 1(b). Average Number of Words? As more words do not strongly correlate with requiring stronger temporal reasoning, how are the authors trying to differentiate TemporalBench from other existing benchmarks?
2. Besides deriving “pull” from “push”, and “left to right” from “right to left” as they appear to be antonymous and natural to think of, how are the rest of the options derived from the correct answer without introducing prejudice of the “centered” word “pull”?

---

### Official Review · Reviewer_RXQd · 2025-11-03

**Soundness:** 2
**Presentation:** 3
**Contribution:** 3
**Rating:** 4
**Confidence:** 4

**Summary:**

The paper proposes TemporalBench, a new benchmark aimed at evaluating fine-grained temporal dynamics understanding in multimodal video models. The authors argue that most existing video QA and video understanding benchmarks can be solved from a single frame or even purely from text, due to coarse annotations that describe high-level actions (“cooking”, “playing guitar”) rather than temporal structure. TemporalBench is built from ~2K videos collected from multiple existing datasets (ActivityNet Captions, Charades, COIN, EgoExo4D, movie clips, Oops, FineGym), and augmented with dense, human-refined temporal captions that describe precise action sequences, frequencies, durations, and temporal relations. This produces ~15K QA pairs.

**Strengths:**

1. Clear and compelling motivation

The paper articulates the single-frame / text-only bias problem in existing video benchmarks convincingly, citing prior work showing that models can often solve them with a single frame or even without visual input. TemporalBench is explicitly designed to counter this by making temporal details the core of the task. The argument that coarse annotations lead to pseudo-video tasks that are effectively static-image recognition is well made and timely.

2. Well-designed benchmark with fine-grained temporal focus

The dataset design targets temporal dynamics explicitly: action frequency (“two vs three slices”), sequence (“push glasses, then drink” vs “drink, then push glasses”), motion direction, and effector changes (right vs left hand), which cannot be reliably guessed from a single frame. The dense captions (≈42 words per clip on average, significantly more words per second than existing benchmarks) support more fine-grained tasks than typical datasets like MSRVTT or TGIF.

**Weaknesses:**

1. My major concern is how the long videos (e.g. 20 mins) are annotated. Does it keep the temporal dynamic over a long span of the video? If it has, how are the long-time dependencies annotated?

2. While ~2K videos and ~15K QA pairs are reasonable for a benchmark, this is still modest compared to many modern video datasets. It would be useful to more explicitly discuss how representative 2K videos are of the wider space of temporal phenomena.

3. The manuscript seems do not provide quantitative measures of annotation quality, such as inter-annotator agreement, consistency checks on counts, or error rates on held-out verification sets. Given that many questions hinge on very fine distinctions (e.g., two vs three swings, direction “corner to center” vs “center to corner”), even small annotation inaccuracies could significantly affect scores.

**Questions:**

Please refer to weaknesses.

---

### Official Review · Reviewer_ytQq · 2025-11-08

**Soundness:** 3
**Presentation:** 3
**Contribution:** 3
**Rating:** 4
**Confidence:** 4

**Summary:**

This paper introduces TemporalBench, a new benchmark dataset designed to evaluate fine-grained temporal understanding in multimodal video models. The authors argue that existing video benchmarks suffer from a single-frame bias, where models can answer questions correctly using static image features without true temporal reasoning. To address this, TemporalBench provides video question-answer pairs that capture complex temporal dynamics like action frequency, order, and motion. The task requires models to distinguish a correct, fine-grained caption from a set of hard negative captions. The paper also identifies a pitfall in standard multi-choice QA and proposes a new metric, Multiple Binary Accuracy (MBA), to provide a more challenging evaluation. The results show a large gap between SOTA models and human performance.

**Strengths:**

1. The paper provides a temporal understanding benchmark with meticulous data curation. The authors used a two-stage human-in-the-loop process for the positive captions.
2. Overall the paper is easy to follow.

**Weaknesses:**

1. The challenges and motivations of this paper are similar to previous work, such as "MVBench: A Comprehensive Multi-modal Video Understanding Benchmark" and "Omnia de EgoTempo: Benchmarking Temporal Understanding of Multi-Modal LLMs in Egocentric Videos". They share the common goal of creating a benchmark that forces the models to infer based on more than a single static frame. The counting problem and action order problem listed in the core contribution have been discussed in previous papers.
2. The proposed evaluation metric is not very well motivated. Please refer to the "Questions" section for more information.
3.  Typo: " to address thi issue" shall be "to address this issue"

**Questions:**

I am willing to raise my score depending on the answers to the following questions.
1. The motivation for the proposed metric is not very strong. Even if the "centralized" pitfall exists, a simple solution is to discard the N(c), and none of those candidate solutions will be preferred. Can you elaborate on the necessity for this metric?

2. As shown in Table 8, the text-only GPT-4o (0 frames) achieves 67.7% BA. Does this high score indicate potential textual bias in your dataset? Could you elaborate on this finding?

3. Can you provide the performance comparison of models other than GPT-4o under the settings of "Text + 1 frame as Input" and "Text + Multi-Frames as Input", respectively?

---

### Official Review · Reviewer_TWji · 2025-11-11

**Soundness:** 3
**Presentation:** 3
**Contribution:** 3
**Rating:** 6
**Confidence:** 3

**Summary:**

TemporalBench is a benchmark focused on fine-grained temporal dynamics in video understanding. It contains ~2K videos with human-authored, temporally detailed captions, yielding ~15K QA pairs (≈10K short, ≈5K long). Negatives are crafted as minimal temporal perturbations (word-level and event-level), intended to force models to reason over counts, order, duration, and motion direction—not just single-frame semantics. The authors identify a structural bias in standard MCQ (a “centralized” distractor effect) and propose Multiple Binary Accuracy (MBA)—factorizing each (M+1)-way MCQ into M binary decisions—to reduce option-structure shortcuts. Experiments span generative video LMMs and video embedding models, with a human AMT reference. Results show a substantial gap: e.g., Gemini-2.5-Pro 43.6% MBA on short-video QA vs humans 67.9%, with further drops on long-video tasks and near-chance performance for embedding models.

**Strengths:**

* Clear problem framing. Pinpoints why many “video” benchmarks collapse to static recognition and single-frame shortcuts; motivates a benchmark where temporal signals are necessary.

* Fine-grained supervision. Human-refined captions capture action frequency, ordering, duration, motion polarity—details that cannot be recovered from a single frame.

* Minimal, targeted negatives. Word-level and event-level edits (e.g., “three slices”→“two slices”) create controlled contrasts that probe specific temporal skills.

* Broad evaluation scope. Covers short (<20s) and long (≤20 min) settings and evaluates both generative and embedding model families with frame-budget studies.

* Metric contribution (MBA). Identifies an MCQ option-structure bias and offers a principled factorization into multiple binary decisions; shows large MCQ vs MBA gaps (e.g., 78.7% vs 43.6%).

* Human baseline & worker hygiene. Excludes caption annotators from evaluation; uses onboarding and curation to ensure reliable human reference.

**Weaknesses:**

* MBA comparability & calibration. MBA multiplies per-option accuracies; difficulty depends on M. If M varies across items/splits, raw MBA isn’t directly comparable. The long-video setup fixes the chance (~9.5%), but short-video details on M distribution and normalization are unclear.

* Residual text-pattern cues. Although MBA mitigates centralized-option bias, lexical/semantic artifacts in positives vs. LLM-generated negatives may remain (style, length, rare tokens). A stronger audit of text-only performance per item type would help.

* Negative generation provenance. Negatives are primarily LLM-generated and then author-filtered; this may inject LLM-specific artifacts that certain models can learn to exploit or avoid. Human-authored adversarial negatives or cross-LLM generation ablations would strengthen claims.

* Annotation consistency & reliability. The two-stage AMT→author pipeline is sound, but the paper lacks inter-annotator agreement, edit-rate statistics, and error taxonomies for refined captions (especially beyond FineGym).

* Embedding model diagnosis. “Near-chance due to small embedding size (768–2048)” is speculative; the deficit may stem from temporal aggregation design, training data, or loss formulations. A dimension-controlled ablation is needed.

* Long-video construction bias. Long items are formed by concatenating dense clip captions; correctness hinges on exact sub-caption alignment. This composition could advantage strategies that match local segments rather than truly reasoning over long temporal narratives.

* Scope & coverage. Audio is removed during annotation (good for visual reliance), but the evaluation setup for audio/subtitles isn’t fully specified. If subtitles/text are ever provided at test time, it risks re-introducing non-visual shortcuts.

**Questions:**

* Does the number of negatives per item (M) vary across your short-video set? If so, how do you normalize MBA so scores are comparable across items/models?

* To what extent do lexical/style artifacts drive performance differences between positives and negatives? What are text-only LLM results under MBA, broken down by negative type (word vs. event), domain, and video length?

* How robust are results to the source of negatives (human vs. LLM) and to cross-LLM generation (e.g., train/eval splits with different negative generators)?

* How do you guarantee unique alignment of each dense sub-caption within concatenated long videos, and can models show where the evidence lies (frame indices)?

---

### Note · Authors · 2025-11-14

I have read and agree with the venue's withdrawal policy on behalf of myself and my co-authors.